# Revisit of the Fennoscandian Shield along the UPPLAND seismic profile: competitive velocity models

Tomasz Janik[1], Raimo Lahtinen[2], Monika Bociarska[1], Piotr Środa[1], Dariusz Wójcik[1]

[1]Institute of Geophysics, Polish Academy of Sciences, Ks. Janusza 64, PL-01-452 Warszawa, Poland
[2]Jalavakuja 4 F8, 02760 Espoo, Finland

*Correspondence to*: Monika Bociarska (bociarska@igf.edu.pl)

**Abstract.** The manuscript presents a comprehensive re-analysis of seismic data collected along the UPPLAND profile in the Fennoscandian Shield, focusing on the competitive velocity models for P-wave (Vp) and S-wave (Vs) velocities, as well as the Vp/Vs ratio. The initial data collection was conducted in 2017, and the first interpretation was published by Buntin et al. in 2021. This study reveals that while both the previous and current models exhibit similar velocities up to a depth of approximately 35 km, significant discrepancies arise in the lower crust and upper mantle velocities, as well as the depth of the Moho boundary. The preferred model indicates Vp values of approximately 7.05-7.17 km $s^{-1}$ in the lower crust and 8.05 km $s^{-1}$ in the upper mantle, contrasting with the earlier model's values of 7.25-7.4 km $s^{-1}$ and 8.0-8.5 km $s^{-1}$, respectively. The Moho depth varies between 43-50 km in the new model, compared to 45-52 km in the previous one.

In addition, we present two, possibly overlapping, tectonic interpretations to explain the new model. The main crustal structure has formed during W-vergent crustal stacking at ca. 1.86 Ga, followed by N–S crustal shortening at 1.82–1.80 Ga. The bulging of the high-velocity upper mantle is either related to extension at 1.89–1.87 Ga in a continental back-arc or during extensional magmatism at 1.7/1.8 Ga. The findings highlight the complexities in determining lower crustal and upper mantle properties from ambiguous seismic data and suggest that the interpretations presented may require a more cautious approach, allowing for alternative explanations.

## 1 Introduction

In 2017, a wide-angle reflection/refraction (WARR) profile named UPPLAND, spanning ~540 kilometres, was carried out in central Sweden. The profile traverses 5 tectonic domains of this part of the Fennoscandian Shield with the Bergslagen region as its core, bounded by broad deformation belts in the north and south (Fig. 1a and b). The analysis of the data obtained along the profile (Figs. 2a and b, and Fig. A1a and b in Appendix A) was presented in the article by Buntin et al. (2021). The velocity model (Fig. A2) was calculated, and advanced tectonic and petrological interpretation was also carried out. The great value of the work is comparative litho-geochemistry and velocity analyses for the model, prepared by I. Artiemieva.

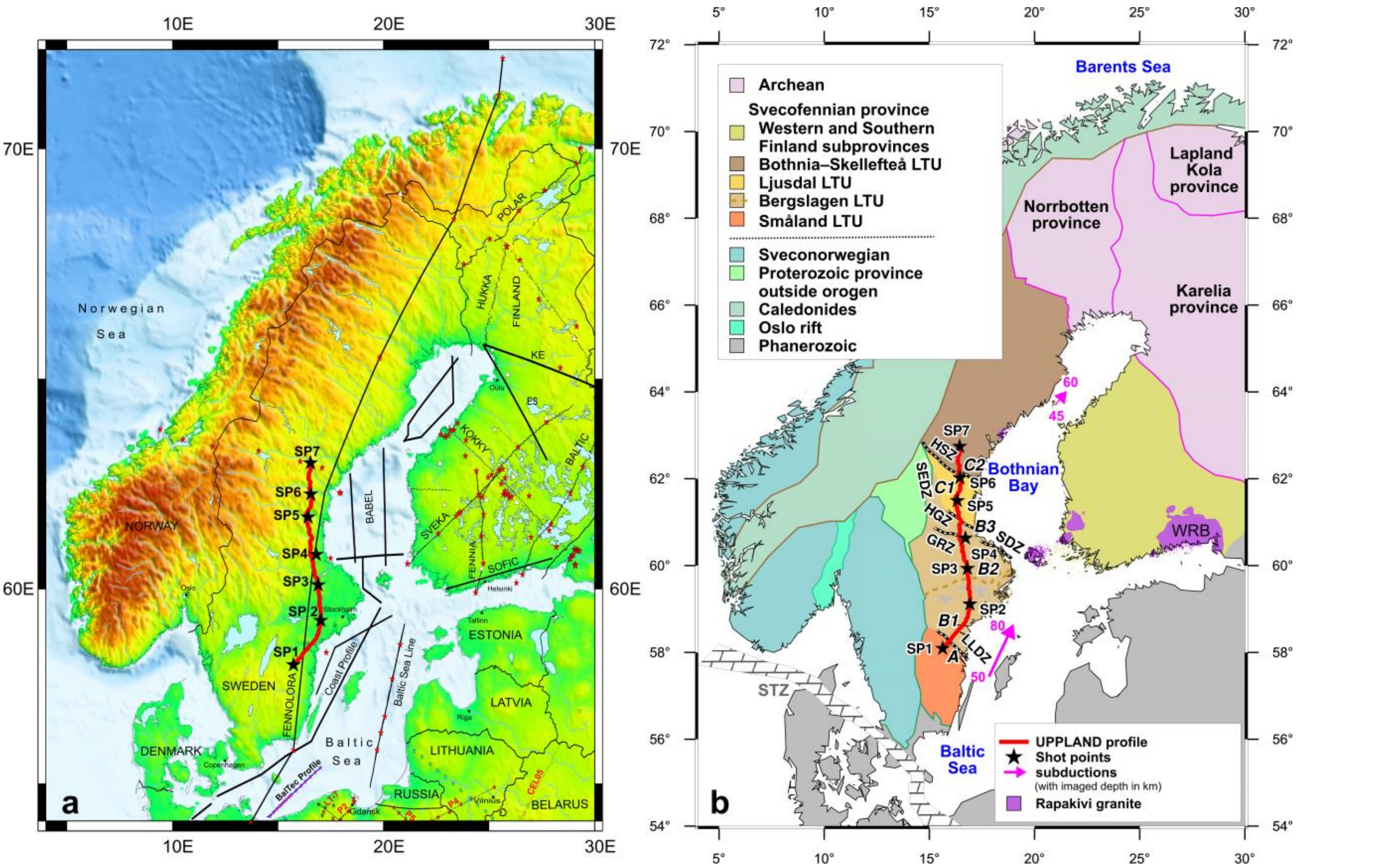

**Figure 1: Location of the UPPLAND profile and previous refraction seismic profiles within the study area (a). Seismic profile superimposed on a tectonic sketch map (modified after Buntin et al., 2021) (b). Tectonic units: A - Småland Terrane, B1 - Sörmland Basin, B2 - Uppland Batholith, B3 - major deformation zone, C1 - Ljusdal Batholith, C2 - Bothnian Basin. Magenta lines - dip of Proterozoic subductions imaged seismically. Lithotectonic units (LTU) after Stephens and Bergman (2020). GRZ - Gävle–Rättvik Zone, HGZ - Hagsta Gneiss Zone, HSZ - Hassela Shear Zone, LLDZ - Linköping–Loftahammar Deformation Zone, SDZ - Singö deformation zone, WRB - Wiborg rapakivi batholith, SEDZ - Storsjön–Edsbyn Deformation Zone, STZ Sorgenfrei–Tornquist Zone. Stars represent shot points, dots receivers.**

After the publication of these results, our alternative model can now be presented. The paper presents the re-analysis of seismic data and calculated tests of the competitive models for the Vp and Vs velocities and the Vp/Vs ratio for the UPPLAND profile. From several seismic models of P- and S-waves velocities and Vp/Vs ratio which fit the travel time data, we selected one and we present it in Figure 3 as our best solution. We will also discuss tectonic interpretations combining existing geological information with the new model.

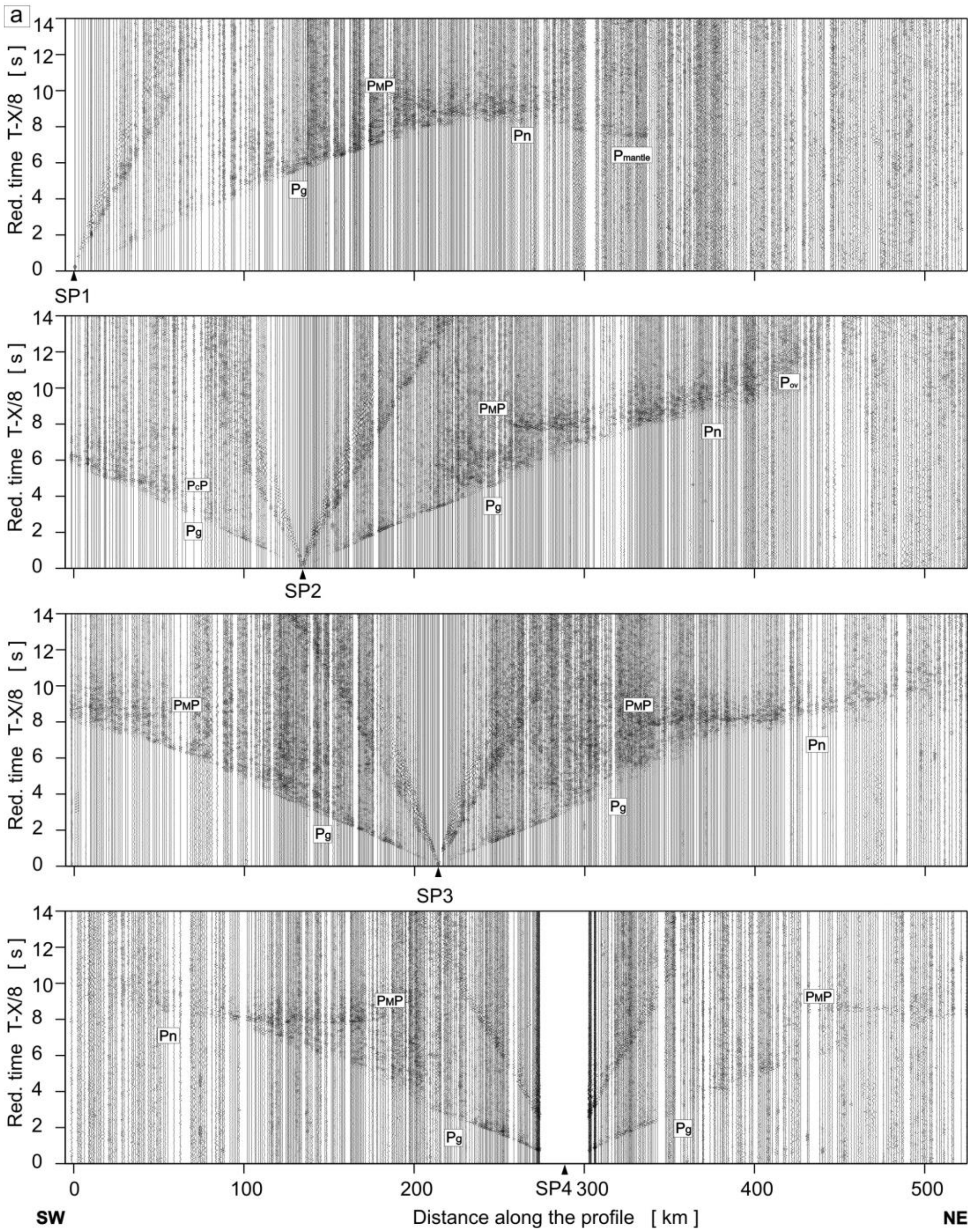

**Figure 2: Examples of trace-normalized, vertical-component seismic record sections for P- and S-wave, SP1-SP4 (a) and SP5-SP7 (b). Band-pass filters, 2-15 and 1-12 Hz, have been applied, respectively. Pg – P refractions from the upper and middle crystalline crust; Pov – P overcritical crustal phases; PcP – P reflections from the mid-crustal discontinuities, $P_MP$ – P reflections from the Moho boundary; Pn – P refractions from the sub-Moho upper mantle; Pmantle – lower lithospheric P phases. Abbreviations have been used for S-wave, respectively. The reduction velocity is 8.0 km $s^{-1}$.**

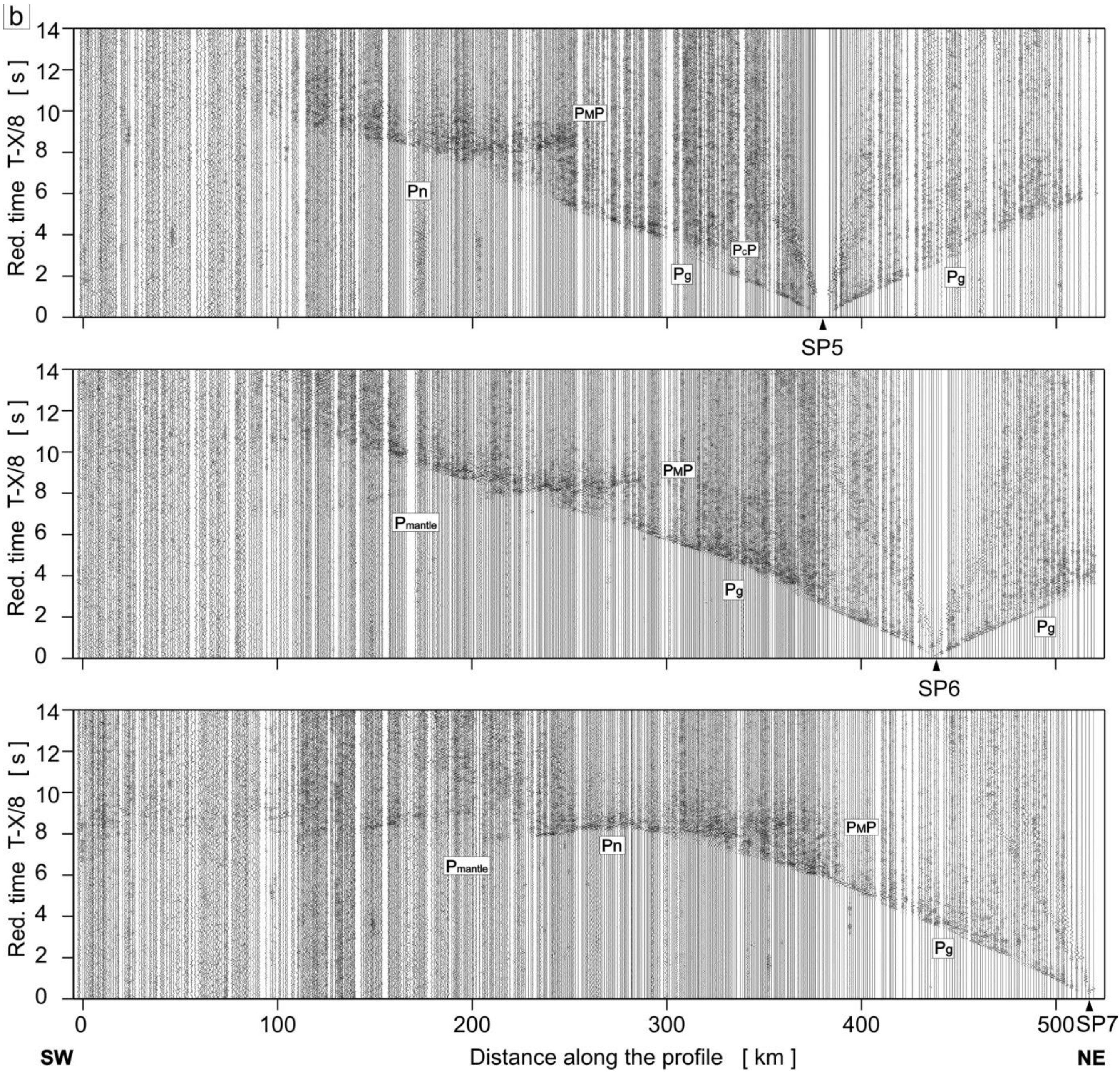

**Figure 2: (continued)**

## 2 Fieldwork and seismic data

The UPPLAND profile is ~540 km long (Fig. 1a). There were seven shot points (SP) located at distances from ~60 km to ~135 km and with charges of 360-500 kg of explosive. Seismic energy was recorded by 595 short-period receivers. For more details, see Buntin et al. (2021) and Fig. A1a and b.

### 2.1 P-wave

The first arrivals of Pg wave are clearly visible at offsets up to approximately ~187-220 km in all recorded seismic sections and in the section for SP7, even up to approximately 260 km (Fig. 2a and b). Apparent velocities (Vapp) vary from ~6 km $s^{-1}$ to ~ 6.75 km $s^{-1}$.

In several sections for further offsets, the apparent velocities of the first arrivals Vapp >7 km $s^{-1}$ are observed (SP1 from ~219 km to ~235 km, Vapp ~7.2 km $s^{-1}$; SP2 (right) from ~192 km to ~267 km, Vapp ~7.0-7.1 km $s^{-1}$; SP4 (right) from ~188 km to ~237 km, Vapp ~7.3 km $s^{-1}$; SP5 (left) from ~191 km to ~209 km, Vapp ~7.1 km $s^{-1}$). However, due to the relatively high noise, it is not certain whether they really represent first arrivals, or perhaps they represent later arrivals, and the actual first arrivals disappear in the noise.

In several sections (but not in all of them) at large offsets, clear arrivals are visible, which, judging by the apparent velocities, arrive from the upper mantle: SP1 Pn (~235-262 km, Vapp ~8 km $s^{-1}$), and Pmantle = Pn1 (~262-350 km, Vapp ~8.75 km $s^{-1}$); SP3 (right) Pn (~216-253 km, Vapp ~7.75 km $s^{-1}$); SP6 (left) Pn1 (~216-253 km, Vapp ~8.75 km $s^{-1}$); SP7 (left) Pn (~231-254 km, Vapp ~8 km $s^{-1}$), Pn1 (~254-300 km, Vapp ~8.75 km $s^{-1}$).

### 2.2 S-wave

The S-waves sections also are of good quality (Fig. A1a and b). However, the first appearances of Sg waves are not clearly visible at offsets up to approximately ~156-257 km, depending on the size of the charges used to excite the energy. Apparent velocities vary from ~3.5 km $s^{-1}$ to ~3.75 km $s^{-1}$.

For the two sections with the highest charges, clear pulses with high apparent velocities are visible, probably coming from the upper mantle: SP1 Sn (~241-261 km, Vapp ~4.5 km $s^{-1}$), and $S_n$1 (~261-368 km, Vapp ~ 5 km $s^{-1}$); SP7 $S_{n1}$ (~257-283 km, Vapp ~5 km $s^{-1}$).

## 3 Seismic modelling

### 3.1 Trial-and-error iterative forward modelling

The modelling of travel-times, rays, and synthetic seismograms was performed using the SEIS83 package (Červený and Pšenčík, 1984) with support from the programs MODEL (Komminaho, 1998) and ZPLOT package (Zelt, 1994) with modifications by Środa. Our calculated Vp model (Fig. 3) for the upper and middle crust seems quite unambiguous, althoug at first glance, it seems clearly different from the model of Buntin et al. (2021) and Fig. A2. Both models, despite quite significant differences in terms of the geometry of the boundaries, present similar velocities Vp and Vs (±0.1 km $s^{-1}$) down to a depth of ~35 km. The main differences concern the velocities in the lower crust and the upper mantle and the depth of the Moho boundary. Determining the velocity in the lower crust from seismic sections is often problematic as the lower crustal refractions typically show in the seismic section as later arrivals and may easily be obscured by the first arrivals' coda.

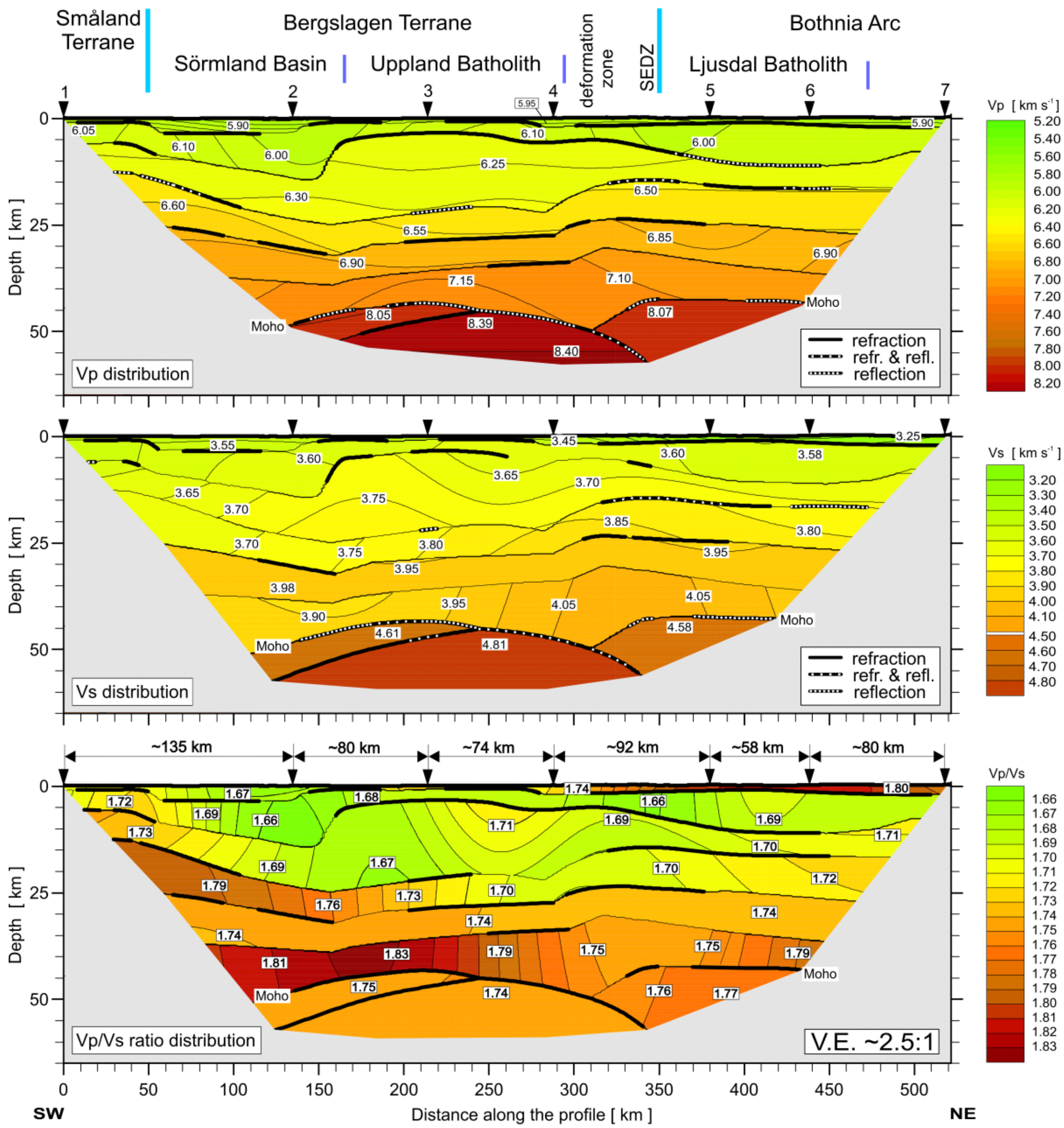

**Figure 3: Two-dimensional models of seismic P- and S-wave velocity in the crust and upper mantle derived by forward ray-tracing modelling using the SEIS83 package (Červený and Pšenčík, 1984) along the UPPLAND profile. (top) P-wave velocity model. Thick, black lines represent major velocity discontinuities (interfaces). (middle) S-wave velocity model. (bottom) Model of Vp/Vs ratio distribution. Thick, black solid and dashed lines represent major velocity discontinuities (boundaries). Only those parts of the discontinuities that have been constrained by reflected or refracted arrivals of P- or S-waves are shown: solid line – refraction only; dashed line – refraction and reflection; dotted line – reflection only. Thinner lines represent inferred velocity isolines with values in km s$^{-1}$ shown in white boxes. The positions of tectonic units at the surface are indicated. Inverted triangles show the positions of shot points. Vertical exaggeration is ~2.5:1 for the model.**

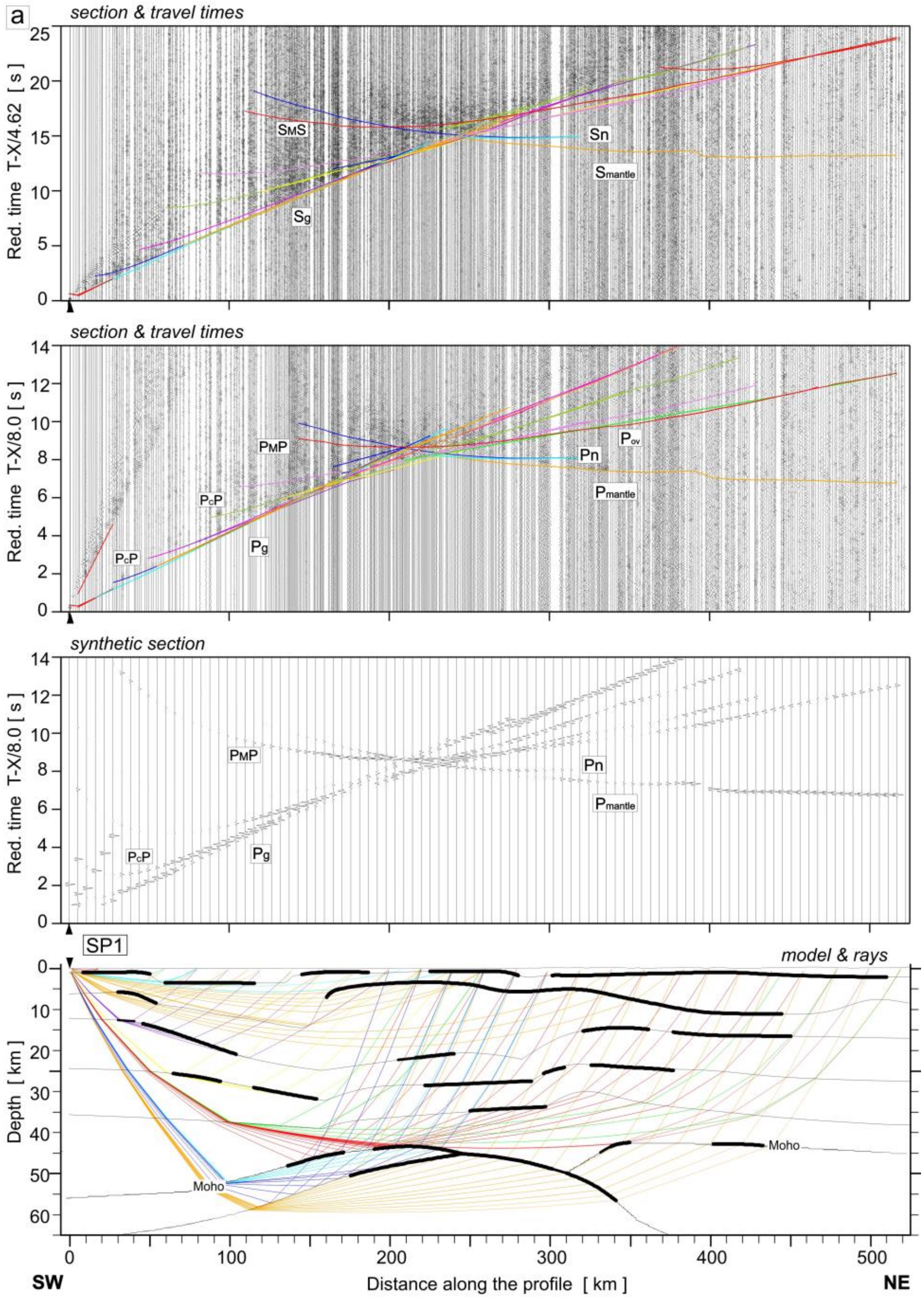

**Figure 4: Examples of seismic modelling along the UPPLAND profile, for SP1 (a) and SP7 (b); seismic record sections (amplitude-normalized vertical component) of S- and P-wave with theoretical travel times calculated using the SEIS83 ray-tracing technique. (top) For S-wave, we used the band-pass filter of 1–12 Hz and the reduction velocity of 4.62 km s$^{-1}$. (top middle) P-wave data have been filtered using the band-pass filter of 2–15 Hz and displayed using the reduction velocity of 8.0 km s$^{-1}$ for P-wave. (bottom middle) Synthetic seismograms and (bottom) ray diagram of selected rays of P-wave. All examples were calculated for the models presented in Figure 3. Other abbreviations are as in Figure 2.**

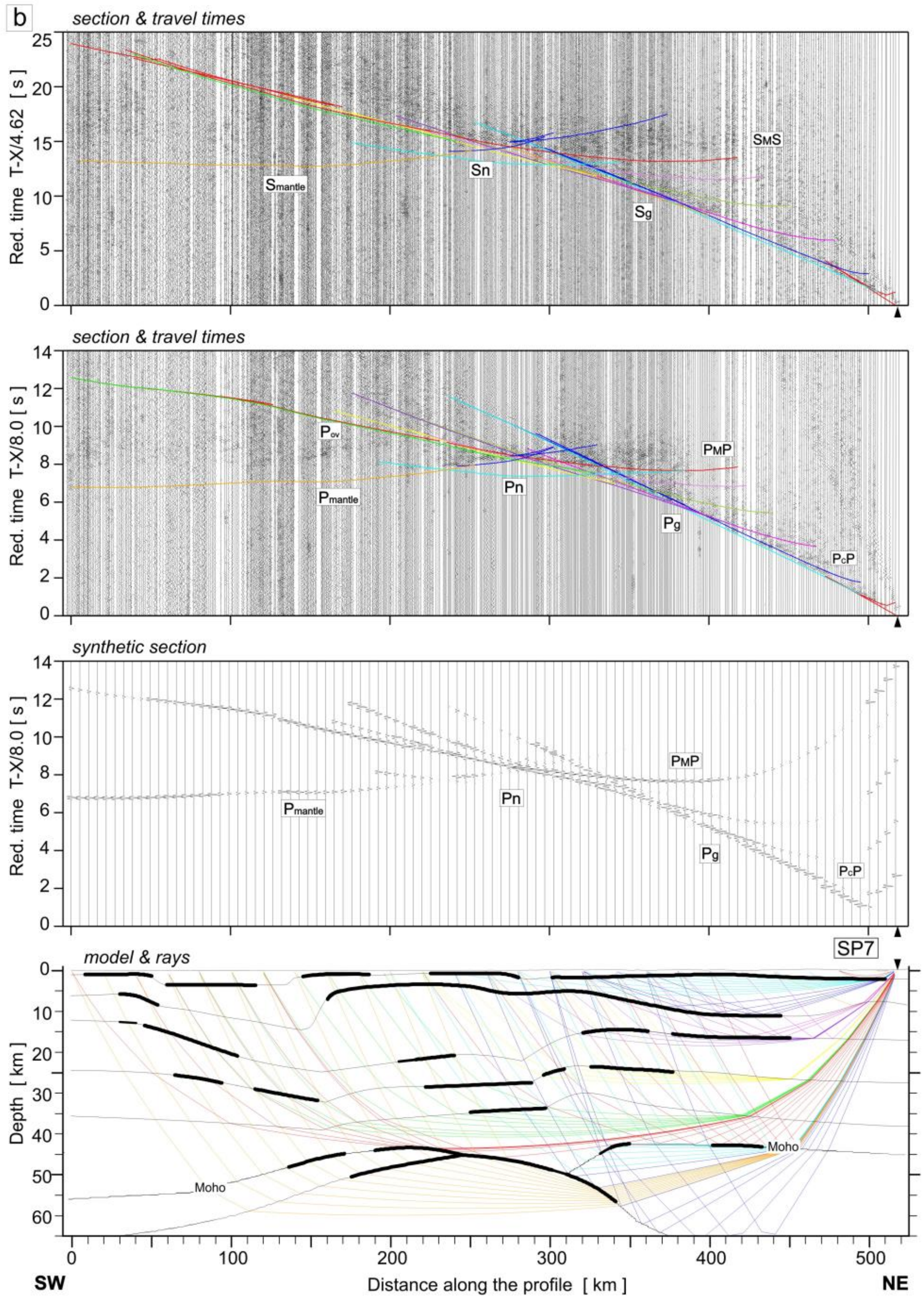

**Figure 4: (continued)**

Using the SEIS83 code, several solutions were tested with different velocities for the lower crust and upper mantle. We conducted tests using three models with different velocities in the lower crust (LC) and upper mantle (UM). Model 1 contained two layers (Vp =7.1-7.15 km $s^{-1}$; Vp ~7.25-7.4 km $s^{-1}$) for LC and Vp ~8.4 km $s^{-1}$ for UM), closely resembling values from

the model of Buntin et al. (2021). Models 2 and 3 had only one layer, Vp ~7.05-7.17 km $s^{-1}$ for LC, and two layers with Vp ~8.05 and Vp ~8.4 km $s^{-1}$ for UM. Models 2 and 3 differ in their UM velocities, particularly in the central part of the profile. Comparing the theoretical and experimental travel times in the seismic sections for P- and S-waves models enabled us to conclude that all three models fall within the class of models acceptable for the specified task, i.e., satisfying available travel time data. However, Model 3, with the best fit (see tests in Appendix A), is our preferred choice. In this model, the depth of the Moho boundary varies in the range of ~44 km (S), ~50 km (central part), and ~42 km (N). The Vp/Vs values in the LC vary from 1.81 through 1.83, 1.75 to 1.79 along the profile from S to N, and for the UM, they range from 1.74 to 1.77. Modelling examples for our Model 3, for both P- and S-waves, are presented in Fig. 4 and Fig. A3a-e. In the Appendix, we present all three tested models (Fig. A4a-c) for Vp, Vs, and Vp/Vs distribution, but their calculated residuals are presented in Tables from A1 to A6.

### 3.2 Uncertainty of the trial-and-error model

The fit of the individual phases of P- and S-waves in Model 3 is shown in Figure 5A and B. At the top (part a), we present the differences between theoretical (black points) and observed (coloured points) travel times. In the middle (part b), we present travel time residuals, and at the bottom (part c), the ray coverage from forward modelling along the profile is shown. We distinguish the Pg arrivals (green points), PcP arrivals (blue points), which are reflections in the crust (without $P_MP$ phase), and $P_MP$ arrivals (red points) and Pn arrivals (brown points). The Pg phase has a good fit along the whole profile in the Model 3. The largest residuals are observed for the $P_MP$ phase in the central part of the profile.

A similar analysis was made for Models 1 and 2, as shown in Figure A5 A-D, respectively. Here, we have also shown differences between theoretical and observed travel times, travel time residuals, and schematic ray coverage from forward modelling along the profile for each model separately. The Pg phase shows a good fit for all shot points. For other phases, in particular for the $P_{M1}P$ phase, the largest residuals are in the middle part of the profile.

For each model, the RMS values were calculated separately for all P-wave and S-wave phases (in Model 1, the Pn1 phase does not occur). The calculated RMS values for each P- and S-phase for all analysed models are shown in Tables from A1 to A6, respectively. The P-wave RMS residuals for Model 1 range from 0.07 s (Pg phase) to 0.23 s ($P_{M1}P$ phase), for Model 2 from 0.07 s (Pg) to 0.19 ($P_{M1}P$), and for Model 3 from 0.07 s (Pg) to 0.13 ($P_MP$). For S-wave, respective residuals are much larger – from 0.28 s to 0.39 s for Model 1, from 0.14 s to 0.26 s for Model 2, and Model 3 gives the lowest S-wave residuals – 0.14 s for Sg phase and 0.18 s for $S_MS$ phase. The total RMS residuals for P- and S-phases for all models are presented in Tables 1 and 2, respectively. Although for different P-waves, the differences between the models are not significant, for S-wave, they are substantial. It can be noticed that Model 3 has the best fit to the data, both for P-waves and S-waves. Additionally, the RMS values for the $P_MP$ and $S_MS$ phases are summarized in Tables 1 and 2, respectively. As before, the RMS differences for the $S_MS$ waves are more significant than for the $P_MP$ waves, and Model 3 shows the best fit to the data.

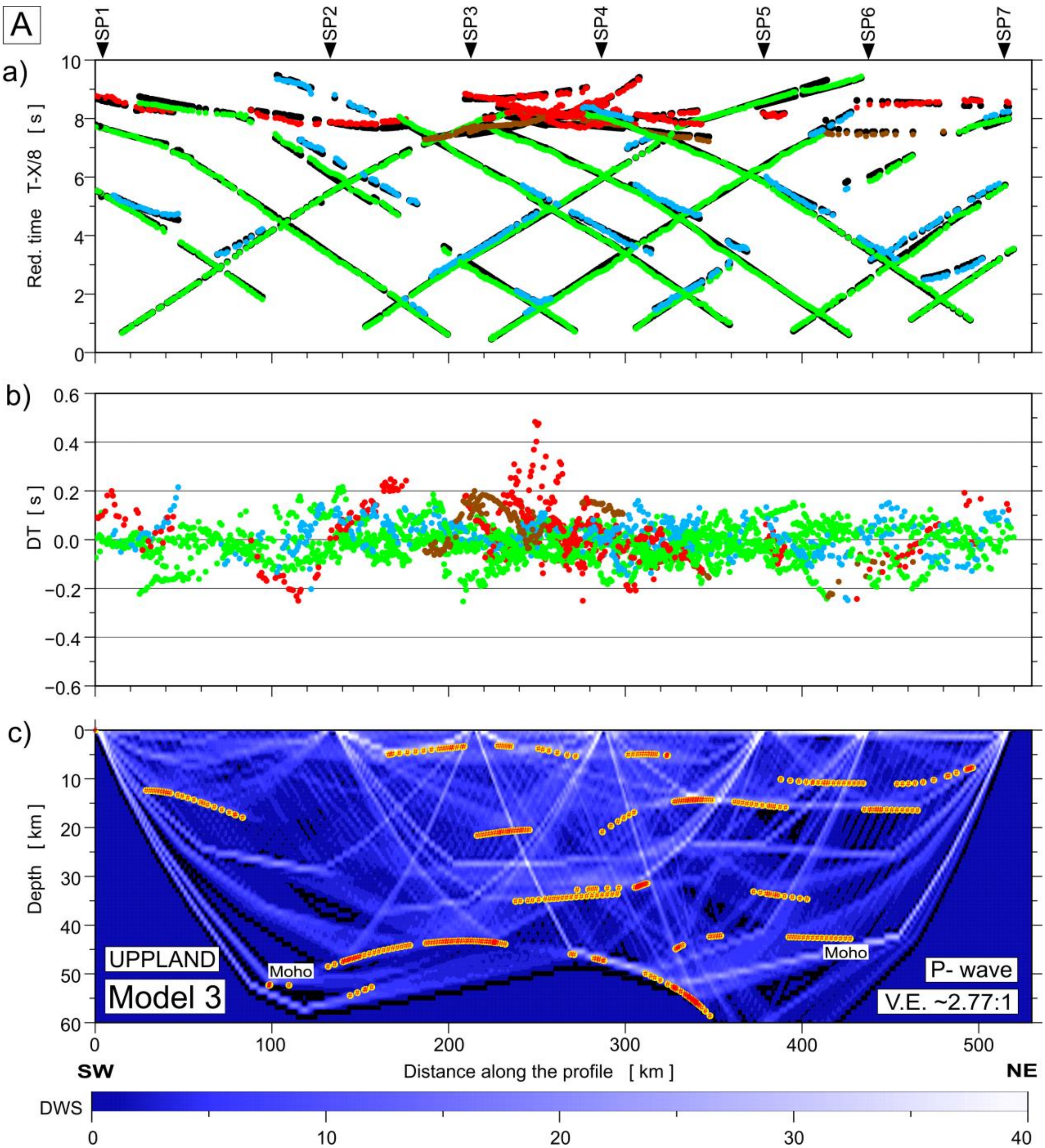

**Figure 5: Diagrams showing theoretical and observed travel times (a), travel time residuals (b), and schematic ray coverage (c) from forward modelling along the profile. Green points – Pg arrivals, blue points – PcP arrivals (reflections in the crust without $P_MP$), red points - $P_MP$, brown points – Pn arrivals, black points – theoretical travel times. Yellow lines – schematic fragments of discontinuities constrained by reflected phases for P-wave velocity Model 3 (A). The red points plotted along the interfaces mark the theoretical bottoming points of reflected phases (every third point is plotted) and their density is a measure of the positioning accuracy of the reflectors. DWS – derivative weight sum. Respective abbreviations have been used for S-wave (B). The reduction velocity is 8.0 km $s^{-1}$ for P-wave and 4.62 km $s^{-1}$ for S-wave.**

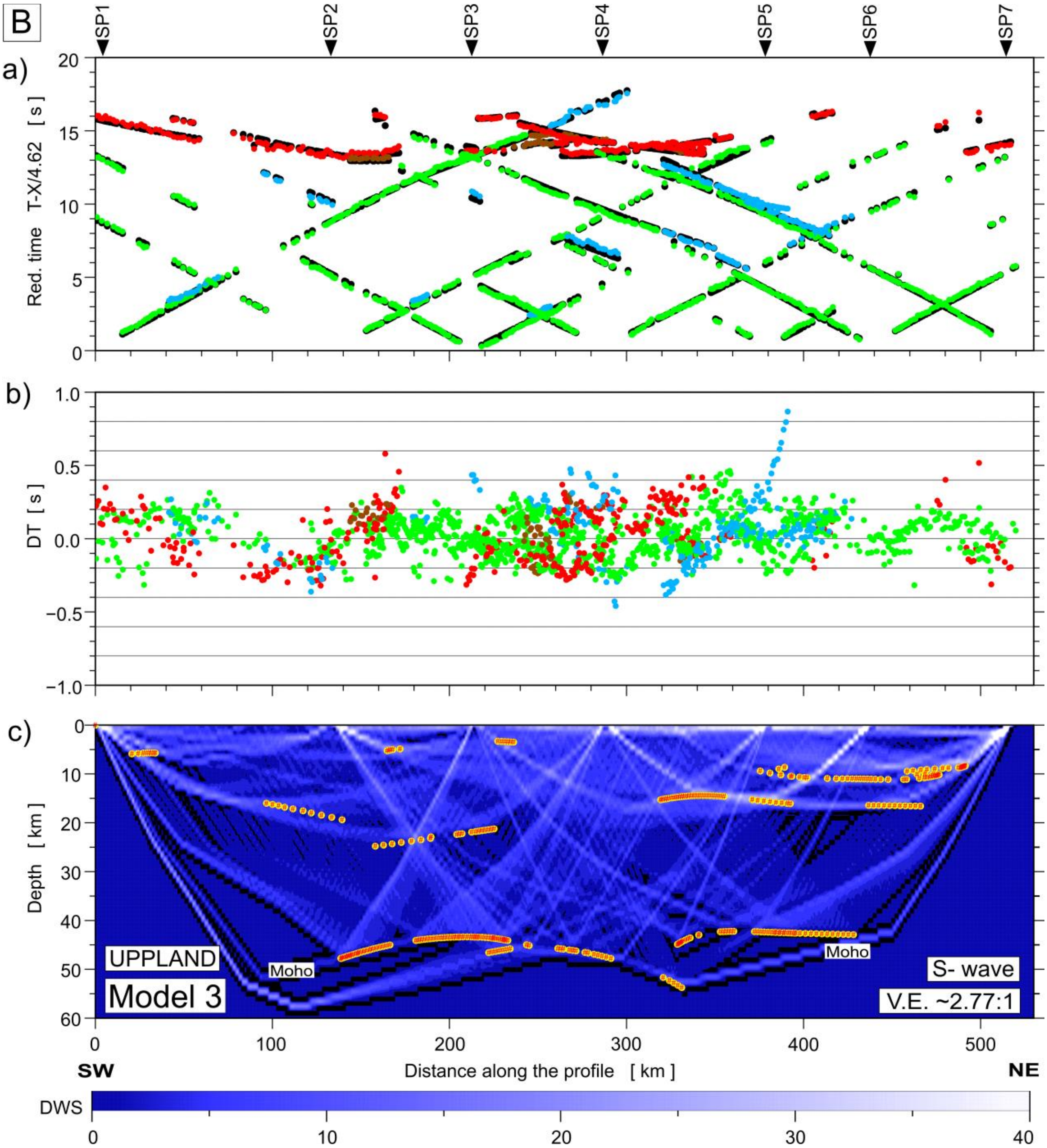

**Figure 5: (continued)**

**Table 1: RMS residuals calculated for the analysed models for both $P_MP$ phase and total values with number of picks, respectively.**

| Model number | Total $P_MP$ RMS [s] | Number of $P_MP$ picks | Total RMS [s] | Total number of picks |
|---|---|---|---|---|
| Model 1 | 0.16 | 500 | 0.09 | 3385 |
| Model 2 | 0.15 | 507 | 0.09 | 3351 |
| Model 3 | 0.12 | 475 | 0.08 | 3312 |

**Table 2: RMS residuals calculated for the analysed models for both $S_MS$ phase and total values with number of picks, respectively.**

| Model number | Total $S_MS$ RMS [s] | Number of $S_MS$ picks | Total RMS [s] | Total number of picks |
|---|---|---|---|---|
| Model 1 | 0.19 | 394 | 0.25 | 1848 |
| Model 2 | 0.26 | 384 | 0.19 | 1724 |
| Model 3 | 0.18 | 379 | 0.16 | 1780 |

### 3.3 Refraction travel time tomography

In order to check which velocity model will be obtained using first arrivals (Psed + Pg + Pn) only and to get an estimate of the non-uniqueness of such model, a tomographic inversion of the P-wave first arrivals of the UPPLAND profile was conducted using the back-projection method proposed by Hole (1992) with various initial models. A 2-D model size of 540 × 70 km was chosen. For preparation of the initial models for the 2-D inversion, first, a 1-D average velocity model was calculated using the Wiechert–Herglotz inversion method. The input for the Wiechert–Herglotz method was an average travel time curve obtained from all first-arrival picks. Then, the mantle velocity was changed to produce three variants of the initial models with mantle Vp of 7.4-7.5 km $s^{-1}$ (Model A), 8.0-8.2 km $s^{-1}$ (Model B), and 8.2-8.4 km $s^{-1}$ (Model C). The first inversion steps were carried out for picks up to 60 km offset, and in subsequent iterations, offsets were increased up to 540 km in 5 steps in order to gradually increase the penetration depth of the seismic rays. In total, 2270 travel times were used in the inversion process. In each iteration, smoothing filters were applied to the velocity corrections. The size of the filter was decreased with iteration number (3 sizes of the smoothing filters were used) in order to gradually increase the resolution. The velocity grid spacing was 1×1 km. For the calculation of the final model, 45 iterations in total were used. The initial models produced RMS residual of 0.17-0.3 s, while the final RMS travel time residual reached for all models was 0.05 s. Considering the estimated picking accuracy as ~0.1 s, the initial RMS residual for some of the models was low, showing that even the initial models show a relatively good fit to the data. This is most likely due to a lateral homogeneity of most of the crust along the profile. The lateral differentiation of the Vp apparently occurs only at lower crustal/upper mantle depths, affecting only a small part of travel times (corresponding to deep Pg rays and Pn rays) and resulting in relatively good overall travel time fit even for 1-D initial models. The final models are presented in Figure A6. It can be seen that in all three models, the main modification of the Vp velocity

field resulting from inversion is the increase of the mantle velocities at a depth of ~45-60 km in the central part of the model, in ~120-340 km distance range, similarly to SEIS'83 forward models.

Other variants of the inversion were done with initial models derived from 2-D SEIS83 raytracing Models 1, 2, and 3. This was done in order to verify the travel time fit of these models to first arrivals data and to check which parts of the models will be modified by the inversion. The result is presented in Figure A7. The RMS residual for these initial models was 0.13-0.20 s (Table A7), close to the estimated picking uncertainty, confirming a good travel time fit for those 2-D raytracing models. Nevertheless, in the final inversion models, we can observe modifications of the Vp distribution, located mainly in the upper mantle of the central part of the model (200-300 km distance) and in the lower crust in its NE part (320-400 km distance). In the effect of the inversion, the high mantle Vp decreased at ~200 km distance and increased at ~300 km distance, shifting the updomed area of high mantle velocities some 70 km to the NE. Also, the lower crustal velocities at ~340-400 km distances were increased by the inversion. However, these changes with respect to the SEIS'83 models may result from using first arrivals only, and SEIS'83 models, as using all refracted and reflected phases, should be considered more reliable. Final RMS residuals after inversion were 0.05 s.

## 4 Discussion

### 4.1 Limitations in wide-angle seismic modelling of the lower crust

We try to describe the problems we encounter when constraining the properties of the lower crust (thickness, seismic wave velocity, heterogeneity) and to discuss the potential limitations of the method. Generally, using wide-angle reflection and refraction, the possible sources of information on the lower crust properties are:

- the first arrivals of the P-wave refracted in the lower crust,
- the Moho reflections at overcritical distances,
- the ringing character of the P-wave signal reflected from or penetrating the lower crust, which gives hints about the fine structure (e.g., lamination) of this layer,
- if observed – S-wave arrivals allow for determination of the Vp/Vs ratio distribution.

Depending on the actual crustal structure and the resulting observed wide-angle wavefield, information about the lower crust can be ambiguous or substantially limited. Non-uniqueness of some parts of the wide-angle model is an inherent feature of this method.

The most appropriate to determine the velocity of the top of the lower crust (LC) are the first arrivals of seismic waves refracted from this boundary. In many cases, the accuracy of the LC modelling can be improved by well recorded overcritical $P_MP$ waves that penetrate the lower crust. However, waves refracted from the lower crust are rarely observed in the first arrivals. In order for them to show up in the first pulses, a sufficiently large LC thickness is needed, as it is, e.g., in the Central Finland Granitoid Complex. Apparent velocities higher than 7 km $s^{-1}$ are seen in most of the record sections at offsets from 185 km up to 290 km profile of the FENNIA and the SVEKA'81 profiles (Janik et al., 2007).

When planning a new wide-angle reflection and refraction profile, the number of shot points must be determined. This depends on the length of the planned profile and the degree of complexity of the crust structure and its expected depth. Considering the high costs of drilling and shooting works, the financial resources available are a major limitation in planning. The optimal distances between adjacent shot points (SP) should be within the range of 1/2H ÷ H (where H – expected crust thickness). Of course, the denser the network of shot points and the better the quality of the recorded sections, the better the final model can represent the complex structure. The distances between SPs on the UPPLAND profile are usually much larger than optimal, as their average is ~90 km. An additional problem is the deviation of the shot points from the straight line of the profile, e.g., for SP1 (~80 km) and, to a lesser extent, for SP5 (~15 km). With such a profile geometry, it is difficult to achieve high model accuracy.

Data from the UPPLAND profile do not enable us to clearly determine the structure of the lower crust. In cases where there is high ambiguity in the measured data, it seems prudent to explore other solutions, particularly for the lower crust. This ambiguity strongly impacts not only the determination of the Moho boundary's depth but also influences the petrological and tectonic interpretation of the studied area. These problems also affect the interpretations discussed below.

## 4.2 Implications for tectonic evolution

The UPPLAND seismic profile (Figs. 1 and 6) crosscuts 1.9–1.8 Ga cratonic crust in south-central Sweden, Fennoscandia. The profile starts from the Småland lithotectonic unit characterized by 1.83–1.82 Ga volcanic arc (OJB) surrounded by 1.81–1.77 Ga granitoids and volcanic rocks (Wahlgren and Stephens, 2020). The Bergslagen lithotectonic unit (Stephens and Jansson, 2020) includes 1.91–1.88 Ga volcanic arc rocks and, in the northern and southern parts, 1.87–1.84 Ga granitoids. The Ljusdal lithotectonic unit is dominated by 1.87–1.84 Ga granitoids (Högdahl and Bergman, 2020) and the profile ends in the Bothnia–Skellefteå lithotectonic unit, including 1.87–1.84 Ga and 1.81–1.77 Ga granitoids (Skyttä et al., 2020). West of these units occurs a large 1.7 Ga magmatic province, which is in part strongly reworked during the Sveconorwegian orogeny (Ripa and Stephens, 2020).

The lithotectonic units are bounded by deformation zones (Fig. 6), which often are kilometers wide zones of gneiss and ductile shear overprinted by localized deformation zones at 1.82–1.80 Ga (for details, see references in Fig. 6). The dextral strike-slip component is dominant, and locally ca. 1.86 Ga older shear deformation is observed. The Storsjön–Edsbyn Deformation Zone (SEDZ) is younger and probably related to the 1.7 Ga magmatic province. The main stages of deformation and metamorphism in the Småland, Bergslagen, and Ljusdal lithotectonic units (for references, see above) occurred at ca. 1.86 Ga (1.87–1.85 Ga) and at 1.82–1.80 Ga (1.84–1.80 Ga). The large regional fold in Bergslagen (Fig. 6a; Stephens, 2020) correlates with the Bergslagen orocline of Beunk and Kuipers (2012). Lahtinen et al. (2023) proposed that Ljusdal and the Mid-Baltic belt of Bogdanova et al. (2015), including the southernmost part of the Bergslagen lithotectonic unit, are parts of a single, originally linear belt characterized by 1.87–1.84 Ga arc magmatism (Fig. 6a).

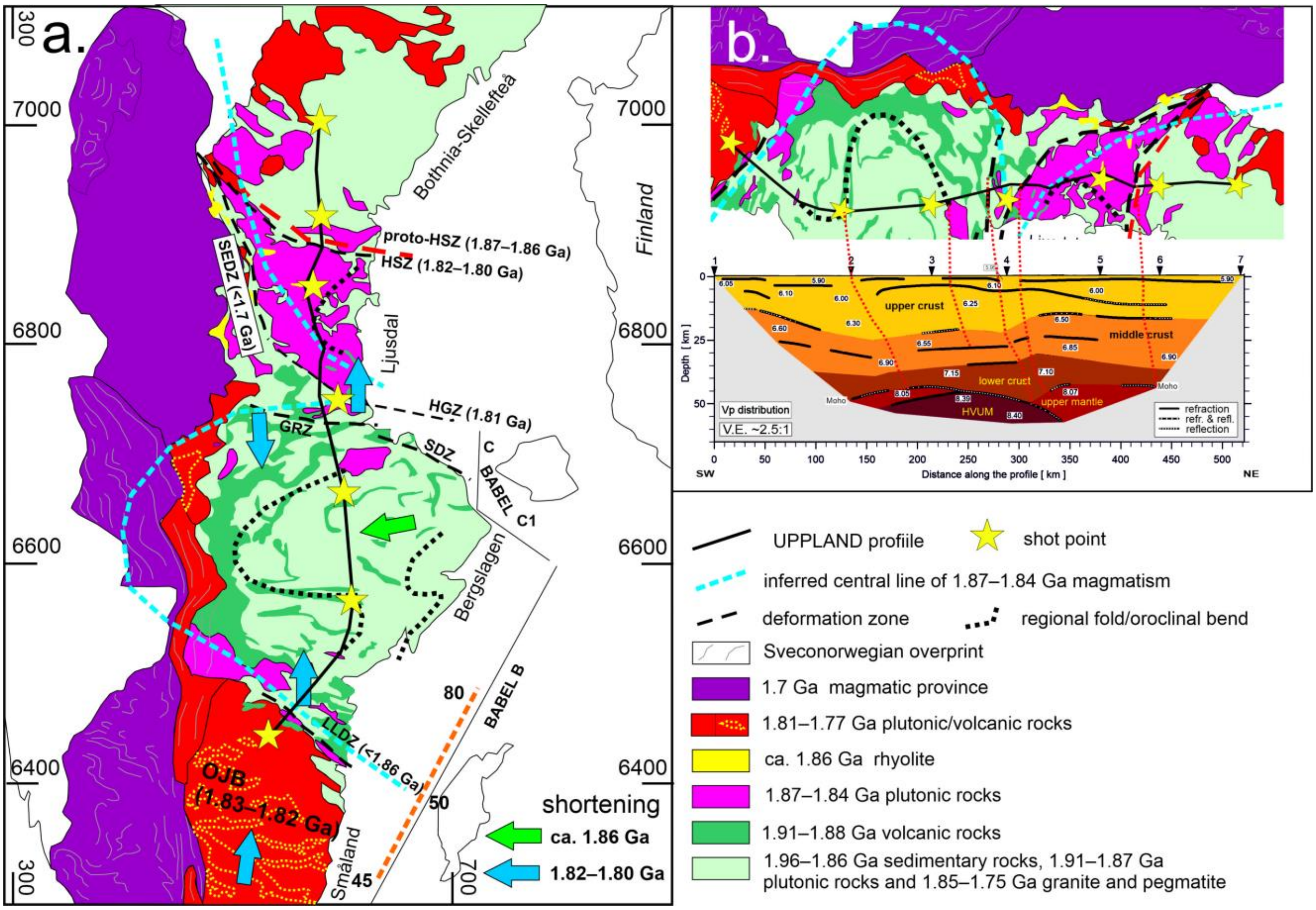

**Figure 6: (a) Simplified geological map of the UPPLAND profile and its surroundings based on descriptions by Högdahl and Bergman (2020); Ripa and Stephens (2020); Skyttä et al. (2020); Stephens and Jansson (2020); and Wahlgren and Stephens (2020). Rocks younger than 1.7 Ga are not shown here, but ≥1.70 Ga rocks reworked in younger orogens are included. The regional folds from Stephens (2020); see Beunk and Kuipers (2012) and ca. 1.86 Ga shortening direction from Stålhös (1981). BABEL profiles (thin lines) and Moho to upper mantle reflector (orange dashed line with imaged depth in km), from Abramovitz et al. (1997). GRZ = Gävle–Rättvik Zone, HGZ = Hagsta Gneiss Zone, HSZ = Hassela Shear Zone, LLDZ = Linköping–Loftahammar Deformation Zone, SDZ = Singö deformation zone, SEDZ = Storsjön–Edsbyn Deformation Zone. OJB = Oskarshamn–Jönköping Belt. (b) Rotated section from the geological map (upper) and modified seismic model along the profile (lower). Dotted red lines show selected geological structures tentatively projected into the seismic model. HVUM = high-velocity upper mantle.**

The tectonic evolution at 1.9–1.8 Ga in the study area has been considered to represent an accretionary orogen in which the crust is stacked sequentially, leading to the lateral growth of the orogen towards the southwest (Gorbatschev and Gaál, 1987; Korja and Heikkinen, 2005; Bogdanova et al., 2015 and references therein). Based on the more generic tectonic switching model by Collins (2002), an episodic evolution containing several extension–contraction cycles in SW-retreating, active continental margin at 1.90–1.80 Ga has been proposed (e.g., Hermansson et al., 2008; Stephens, 2020). Thus, either a single subduction system with intervening crustal shortenings due to flat subduction (e.g., Hermansson et al., 2008) or several subduction-collision pairs (e.g., Korja and Heikkinen, 2005; Buntin et al., 2021) are proposed. The occurrence of similar age (1.91–1.89 Ga) arc rocks in Fennoscandia has been suggested to represent a ≥2000 km long arc system affected by an orocline-

forming event (Lahtinen et al., 2014), which may have included the separation of Bergslagen from the linear arc (Lahtinen et al., 2023). The current structural trends in Bergslagen resulted from the late orogenic, major folding (oroclinal bending) of originally NW–SE oriented structures (Beunk and Kuipers, 2012; Stephens, 2020).

We use the preferred seismic model of this study in our interpretation (Fig. 6b). The proposed paleosubduction zone (Figs. 1 and 6a) could be related to 1.86 Ga and/or 1.82–1.80 Ga flat subduction stages. The N-dipping structures in the upper and middle crust of the seismic model (Fig. 6b) are correlated with N–S crustal shortening (Fig. 6a). The thick upper crust and bulging of the upper mantle (HVUM in Fig. 6b) under Bergslagen occur below the proposed regional fold and the oroclinal bend of 1.87–1.84 Ga magmatism (Fig. 6b). The latter structure ends to a wide deformation zone composed of the GRZ and HGZ (Fig. 6a and b). The depth of the Moho boundary increases abruptly at the end of the BABEL B profile and continues as such in BABEL C1 (Korja and Heikkinen, 2005), correlating with the UPPLAND model (Fig. 6a, b). The Bergslagen–Ljusdal boundary zone is characterized by an upward-expanding and thickening lower crust, as well as bulging of the upper mantle (Fig. 6b). Similar thickening of the lower crust in BABEL C is interpreted as due to N-vergent crustal stacking of the lower crust (Korja and Heikkinen, 2005). We propose a viable tectonic model where the mantle bulge and thinning of the lower crust are related to ca. 1.89–1.87 Ga extension in a back-arc setting of an NW–SE trending continental arc. Subsequent WSW-directed basin inversion at ca. 1.86 Ga was followed by nearly orthogonal shortening at 1.82–1.80 Ga, leading to oroclinal bending and crustal stacking.

The bulging of the high-velocity upper mantle, two upper mantle layers, and lack of high-velocity lower crust in the seismic model (Figs. 3 and 6b) are comparable with the seismic structure under the Wiborg rapakivi batholith (Fig. 1b; Janik, 2010, Tiira et al., 2022). The possibility exists that the above-discussed mantle–lowermost crust structure in the UPPLAND profile has formed during the 1.8 Ga and/or 1.7 Ga magmatic stage(s). Especially, the 1.7 Ga magmatic province seems to have been a very large province, originally extending to the west and possibly also to the east under Bergslagen and Ljusdal (Fig. 6a).

The main problem in the interpretation of the upper to middle crustal structures along the UPPLAND profile is that the ca. 1.86 Ga crustal stacking, preceding the 1.82–1.80 Ga folding and stacking, had vergence towards W, nearly orthogonal to the profile (Fig. 6a). Needed 3-D information would require an E–W oriented seismic profile across Bergslagen. As discussed in this paper, the information about lower crust and Moho boundary can be ambiguous. Also, the age (1.9 Ga, 1.8 Ga, 1.7 Ga or younger) of the lower crust and upper mantle structures is unknown. Based on existing geological information and tectonic models, two possible interpretations of the studied area are discussed above. These models could even be a diachronic process where the high-velocity upper mantle–lowermost crust stabilized at 1.7 Ga, and thus >100 Ma later than the formation of main parts of the crustal structure at 1.82–1.80 Ga.

A contradictory model of solely northward subduction, including a collision between Bergslagen and Ljusdal, was proposed by Buntin et al. (2021). They interpreted the high-velocity body (HVUM in Fig. 6b) as a mafic lowermost crustal layer partially transformed into ca. 150–200 km long and 6–8 km thick eclogite body during Paleoproterozoic orogeny. Interestingly, the high-velocity upper mantle (Vp ~8.30–8.37 km $s^{-1}$) under the Wiborg rapakivi batholith is at least 250 km long (Tiira et al., 2022) but is apparently related to the formation of the rapakivi batholith, occurring >150 Ma later than stabilization of the orogenic crust.

## 5 Conclusions

Our re-analysis of seismic data and the calculated competitive models for the Vp and Vs velocities and the Vp/Vs ratio for the UPPLAND profile show similar velocities Vp and Vs (±0.1 km $s^{-1}$) up to a depth of ~35 km as in the model by Buntin et al. (2021). The main differences between these two models include significant differences in terms of the geometry of the boundaries, the velocities in the lower crust and upper mantle and the depth of the Moho boundary.

Two, possibly overlapping, tectonic interpretations are proposed to explain the new model. The main crustal structure has formed during W-vergent crustal stacking at ca. 1.86 Ga followed by N–S shortening at 1.82–1.80 Ga. The bulging of the high-velocity upper mantle is related to extension at 1.89–1.87 Ga in a continental back-arc or to extensional magmatism at 1.7 Ga and/or 1.8 Ga.

## Acknowledgements

The UPPLAND profile measurement project was sponsored by the Swedish Research Council (VR) (project number 2015-05177). Participation of the Polish group (148 short-period seismic recorders, two cars and 4 people from IG PAS) in this work was supported by a subsidy from the Polish Ministry of Education and Science for the Institute of Geophysics, Polish Academy of Sciences. The authors would like to thank the other project participants who contributed to the data acquisition. The public domain GMT package (Wessel and Smith, 1995) was used to produce some of the figures.

## Author contribution

TJ: Writing – original draft, Visualization, Supervision, Resources, Methodology, Formal analysis, Conceptualization. RL: Writing – original draft, Visualization, Tectonic analysis. MB: Investigation, Writing – original draft, Visualization, Formal analysis. PŚ: Writing – original draft, Visualization, Methodology, Formal analysis. DW: Investigation, Data curation.

## Declaration of competing interest

The authors declare that they have no conflict of interest.

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

## Appendix A

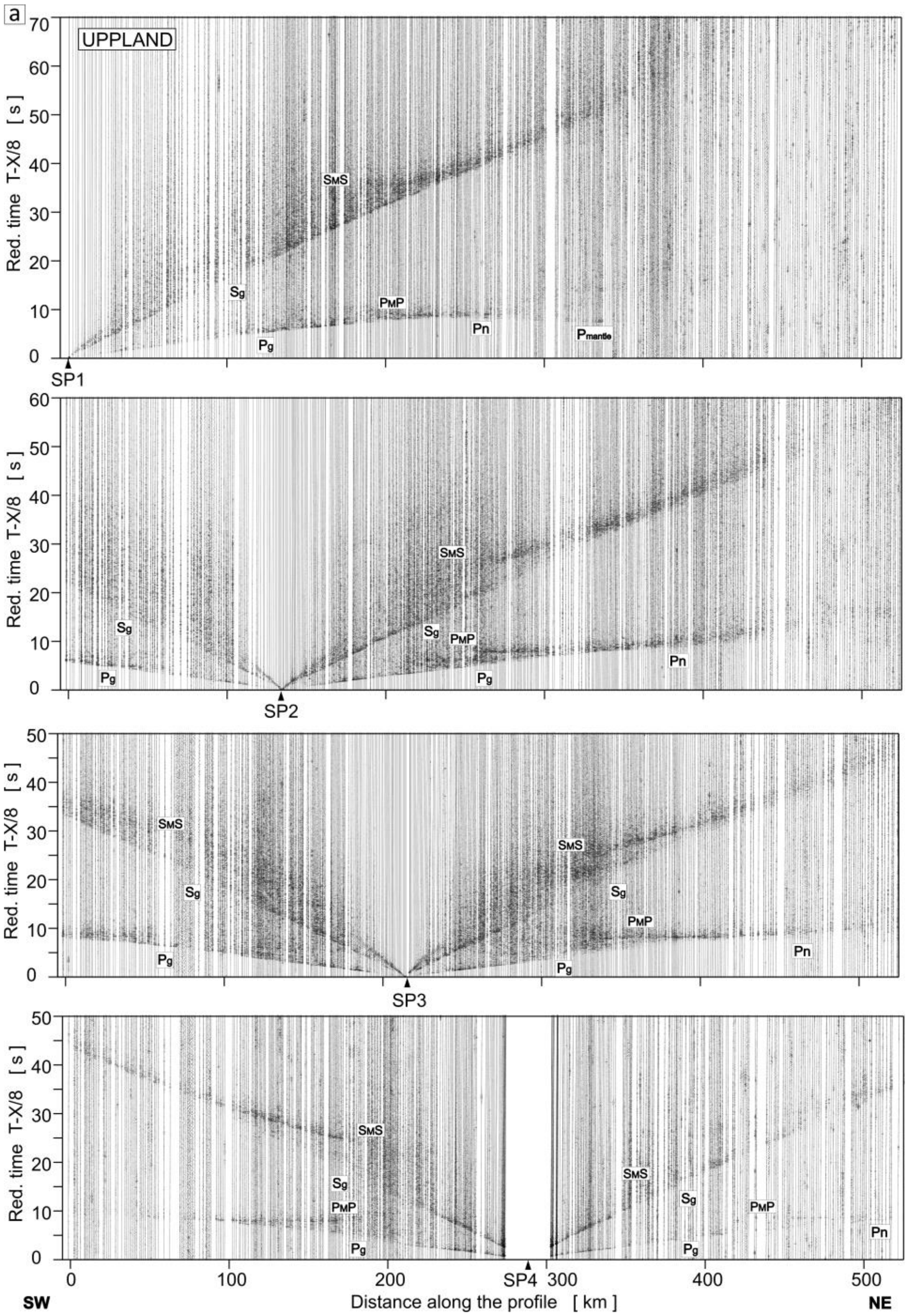

**Figure A1: Examples of trace-normalized, vertical-component seismic record sections for P- and S-wave, SP1-SP4 (a) and SP5-SP7 (b). A band-pass filter of 2-15 Hz has been applied. Psed - P refractions from sedimentary layers; Pg – P refractions from the upper and middle crystalline crust; Pov – P overcritical crustal phases; PcP – P reflections from the mid-crustal discontinuities, $P_MP$ – P reflections from the Moho boundary; Pn – P refractions from the sub-Moho**

**upper mantle; Pmantle – lower lithospheric P phases. Abbreviations have been used for S-wave, respectively. The reduction velocity is 8.0 km s-1.**

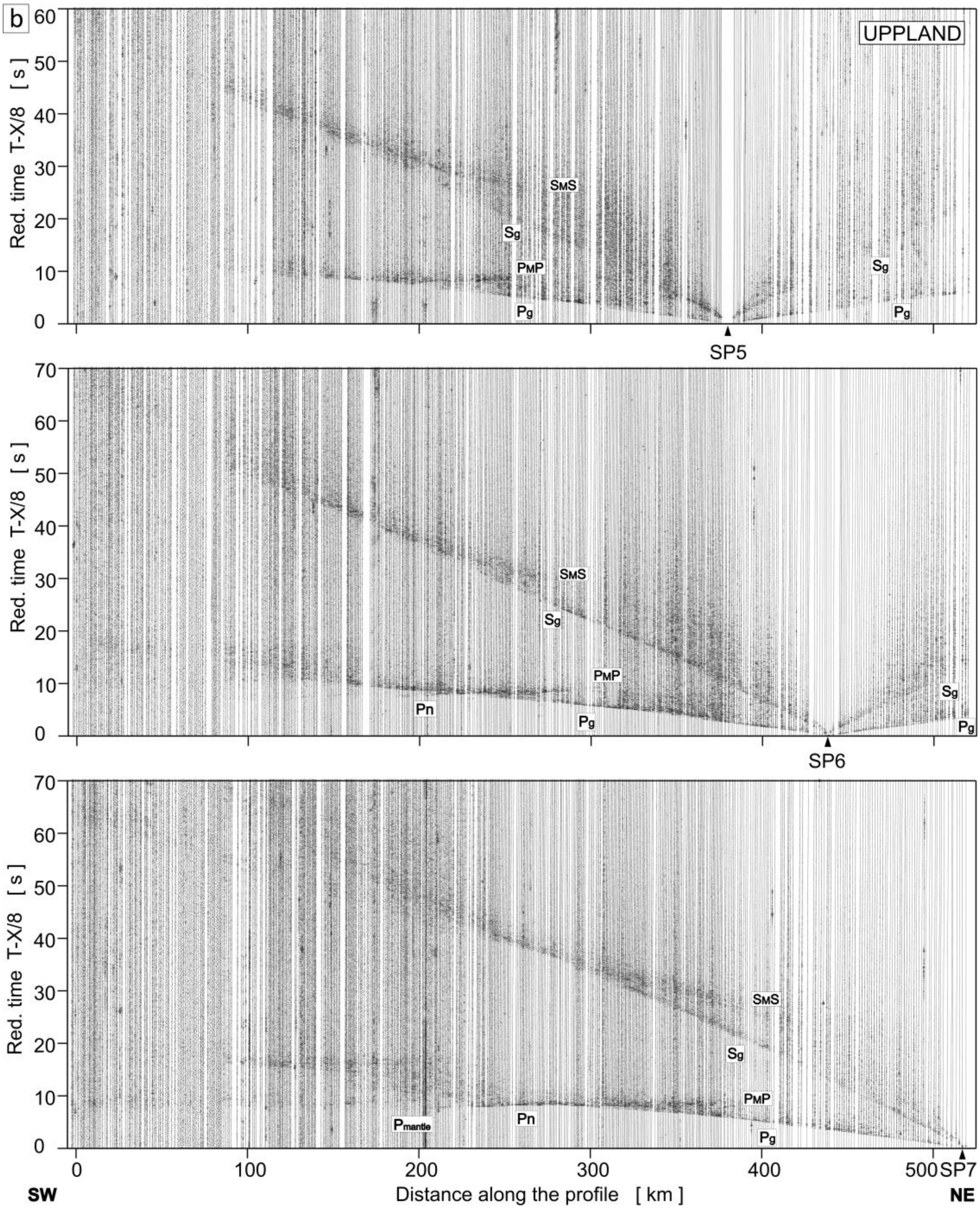

**Figure A1: (continued)**

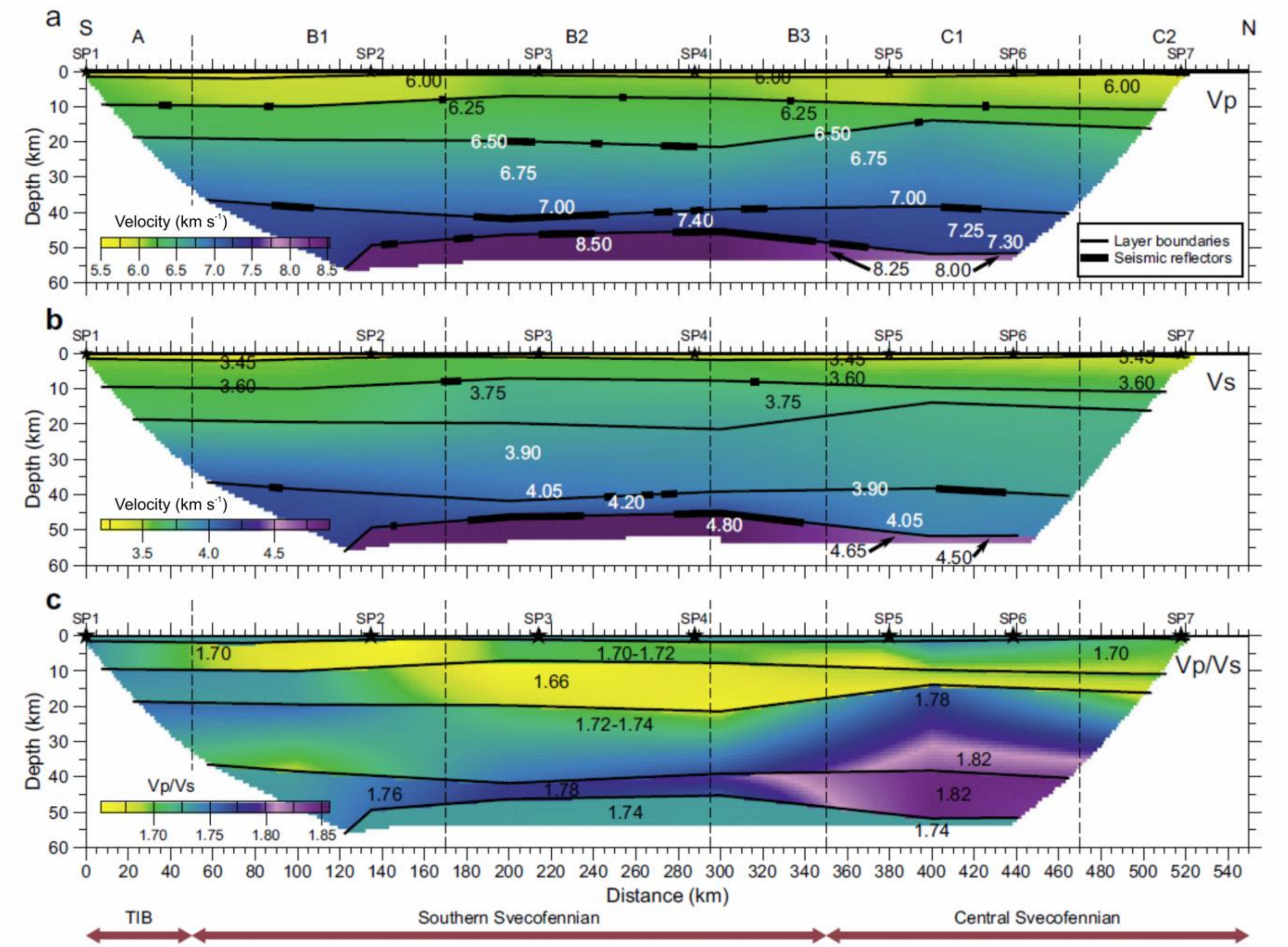

**Figure A2: Seismic models along the profile. (a) P- and (b) S-wave velocity and (c) Vp/Vs ratio along the seismic profile (modified after Buntin et al., 2021). Seismic sources (SP1 - SP7) - black stars; tectonic units as in Figure 1 (main part). Velocity discontinuities - dashed lines; identified seismic reflections - thick black lines (shown in a, b only).**

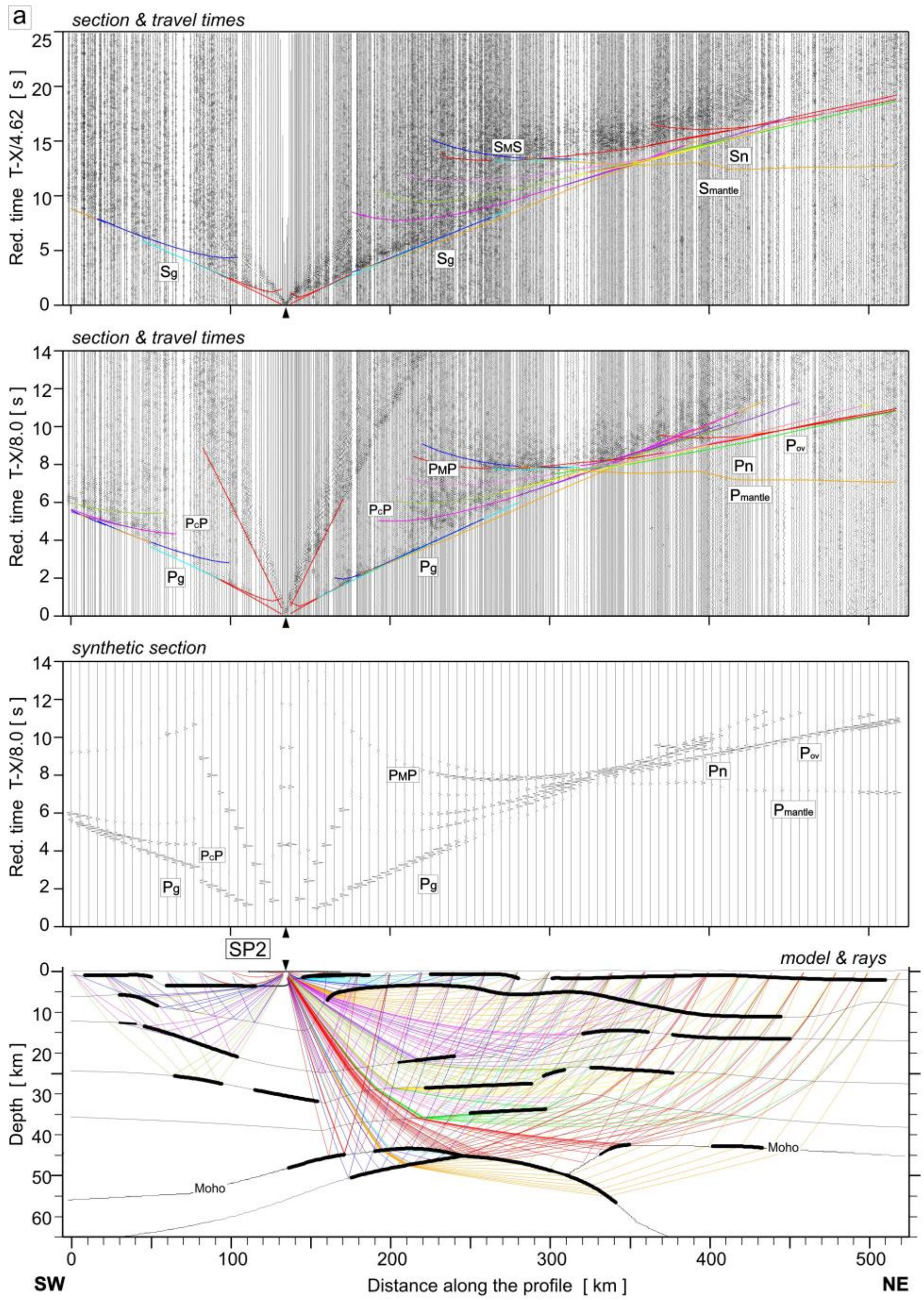

**Figure A3: Examples of seismic modelling along the UPPLAND profile, for SP2 (a), SP3 (b), SP4 (c), SP5 (d), and SP6 (e); seismic record sections (amplitude-normalized vertical component) of S- and P-wave with theoretical travel times calculated using the SEIS83 ray-tracing technique. (top) For S- wave, we used the band-pass filter of 1–12 Hz and the reduction velocity of 4.62 km s$^{-1}$. (top middle) P-wave data have been filtered using the band-pass filter of 2–15 Hz and displayed using the reduction velocity of 8.0 km s$^{-1}$ for P-wave. (bottom middle) Synthetic seismograms and (bottom) ray diagram of selected rays of P-wave. All examples were calculated for the models presented in Figure 3 (main part). Other abbreviations are as in Figure A1.**

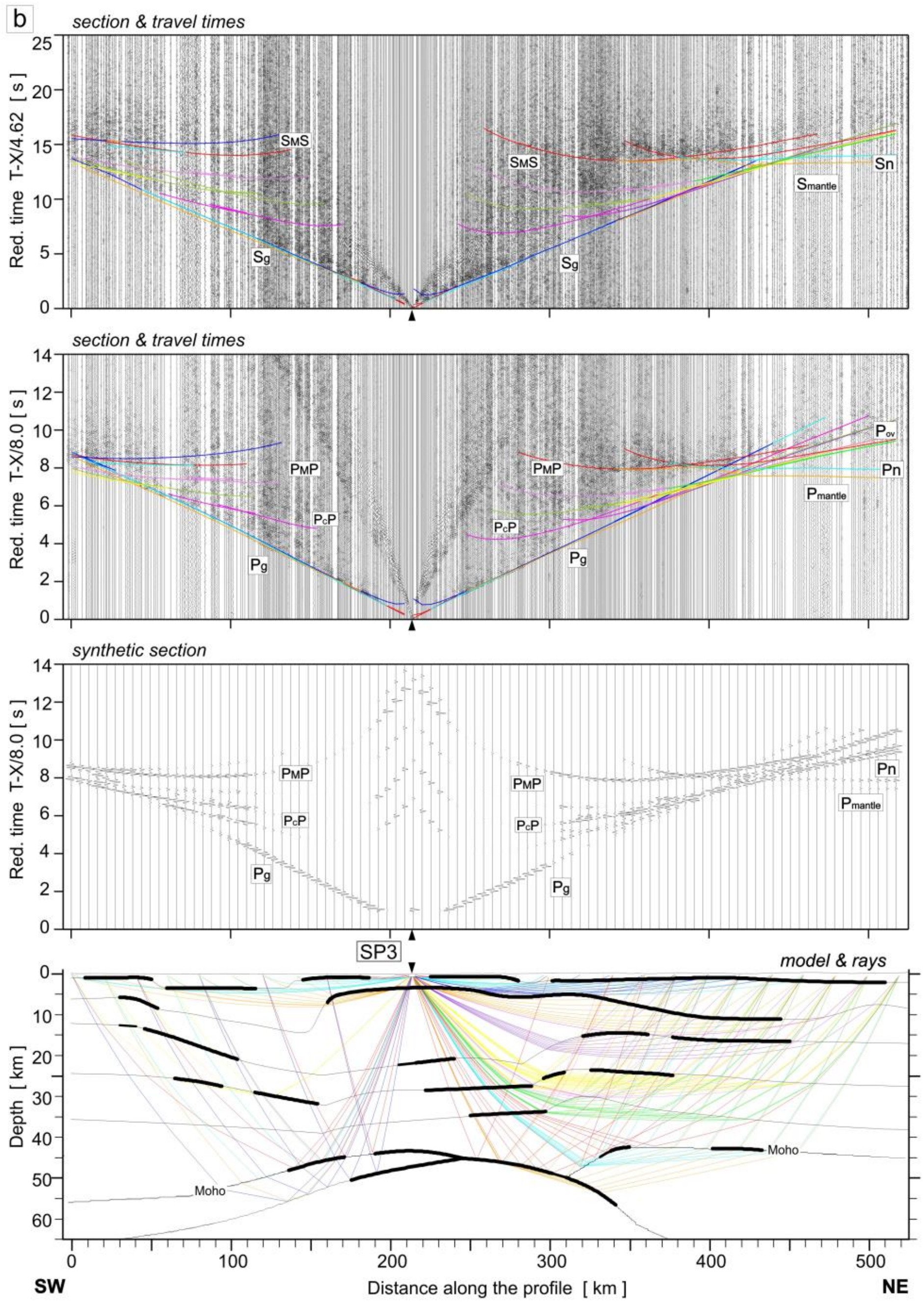

**Figure A3: (continued)**

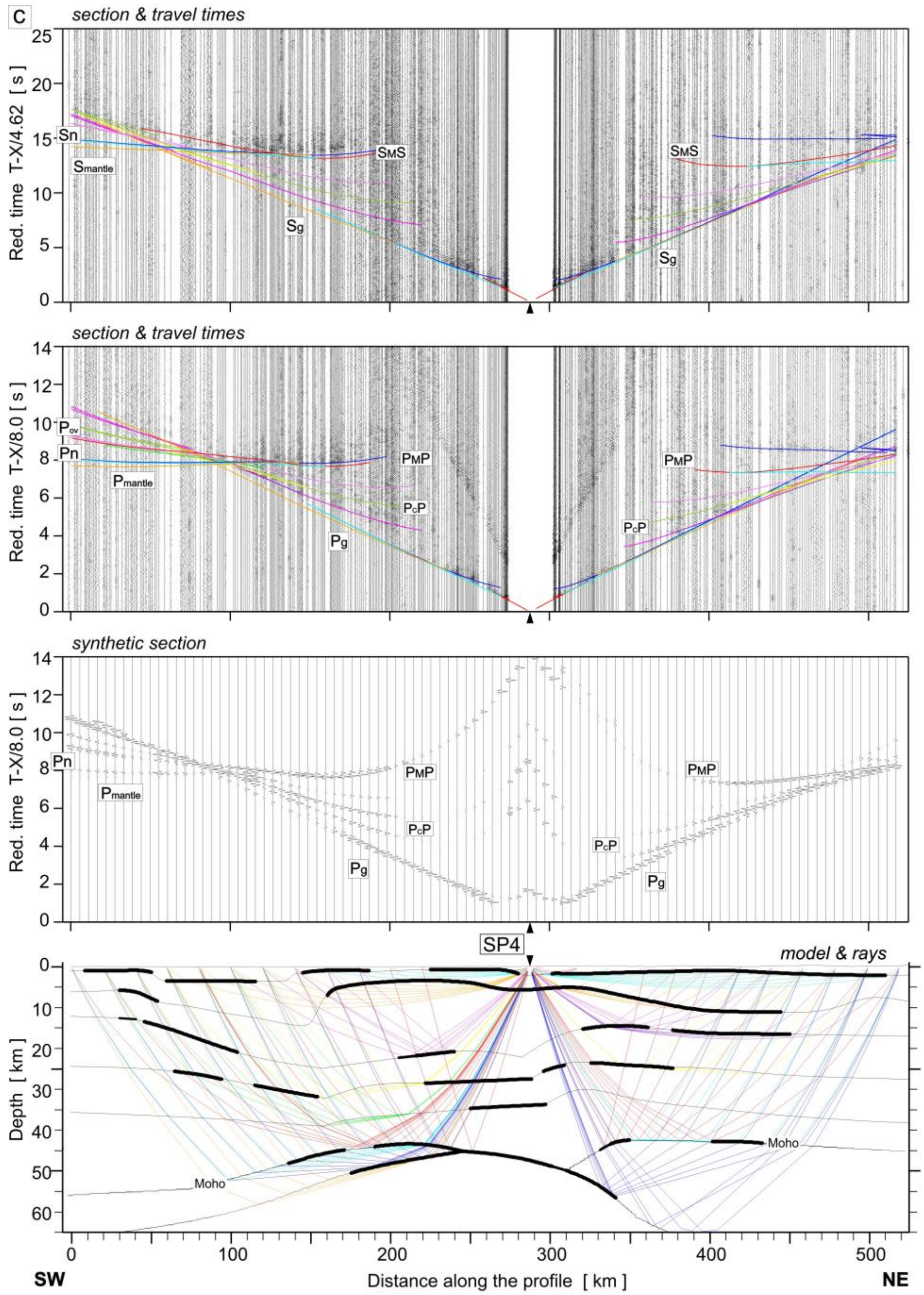

**Figure A3: (continued)**

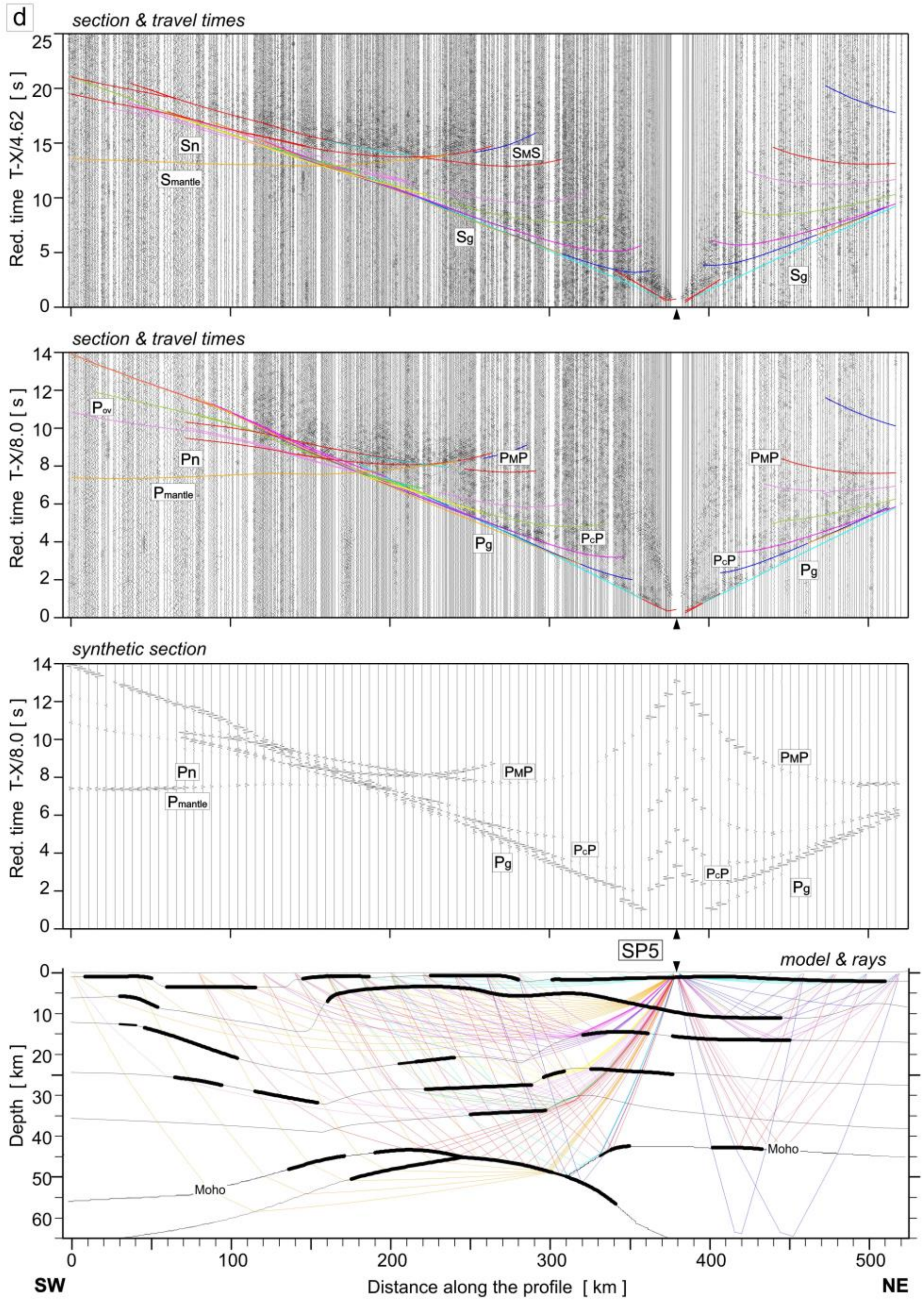

**Figure A3: (continued)**

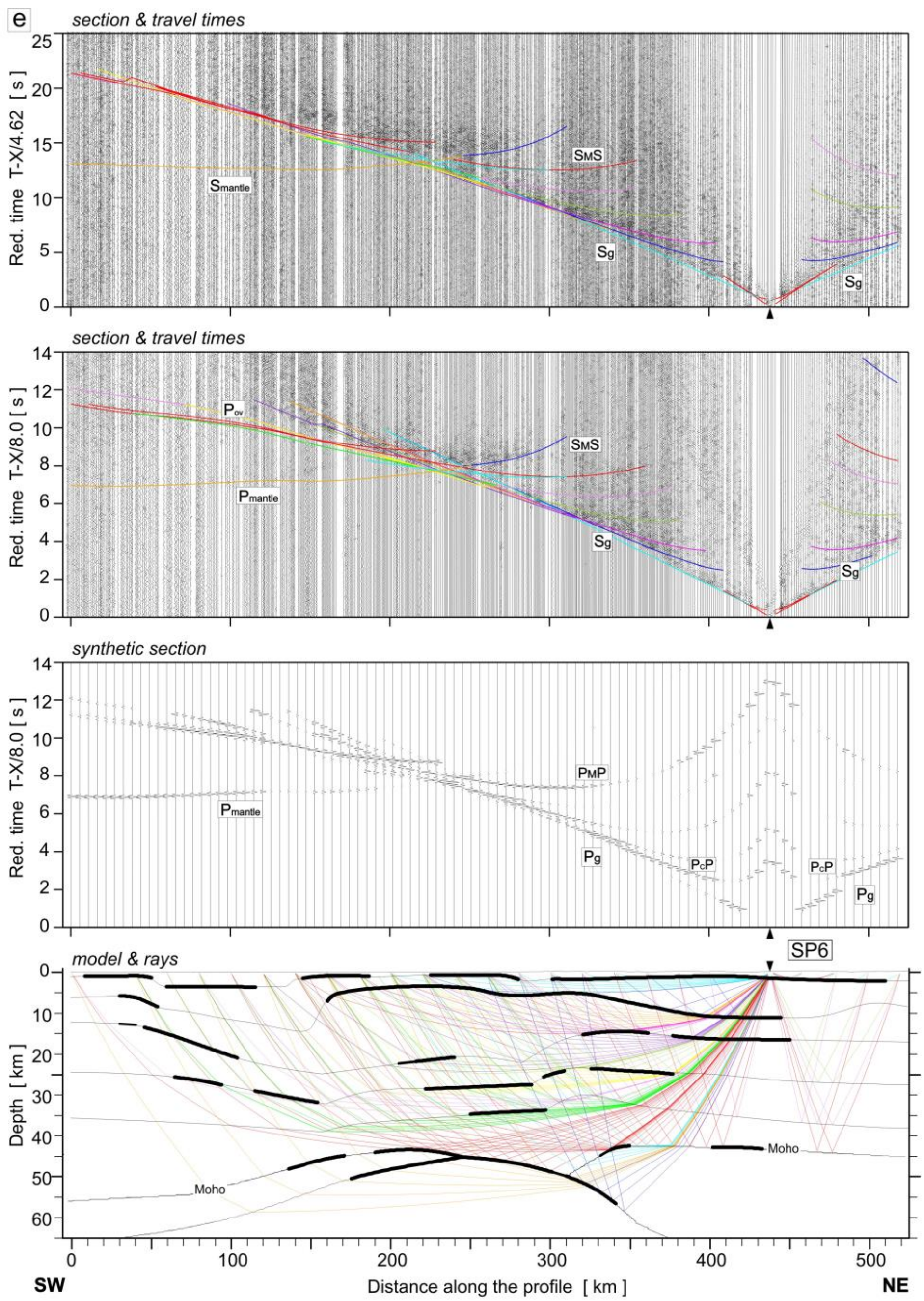

**Figure A3: (continued)**

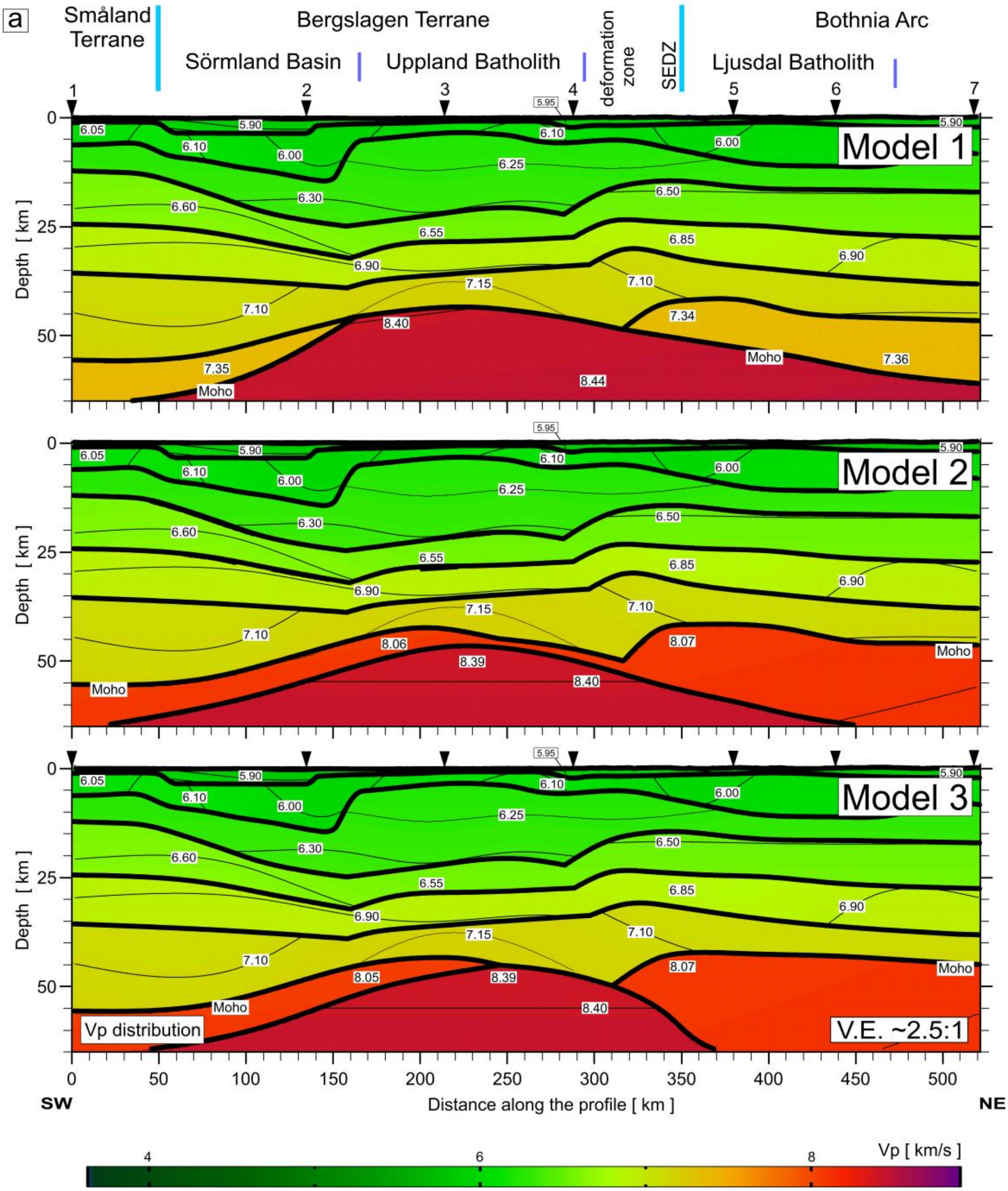

**Figure A4: Tests for 3 models: Model 1, Model 2, and Model 3, differing in boundary geometry and velocities in LC and UM (differences described in the text). Two-dimensional models of seismic P- and S-wave velocity in the crust and upper mantle derived by forward ray-tracing modelling using the SEIS83 package (Červený and Pšenčík, 1984) along the UPPLAND profile: (a) P wave velocity models; (b) S wave velocity models; (c) Model of Vp/Vs ratio distribution. Thick, black lines represent major velocity discontinuities (interfaces). Thin lines represent velocity isolines with values in km s$^{-1}$ shown in white boxes. The position of large-scale crustal blocks is indicated (after Buntin et al., 2021). Arrows show the positions of shot points. Vertical exaggeration is ~2.5:1 for the models.**

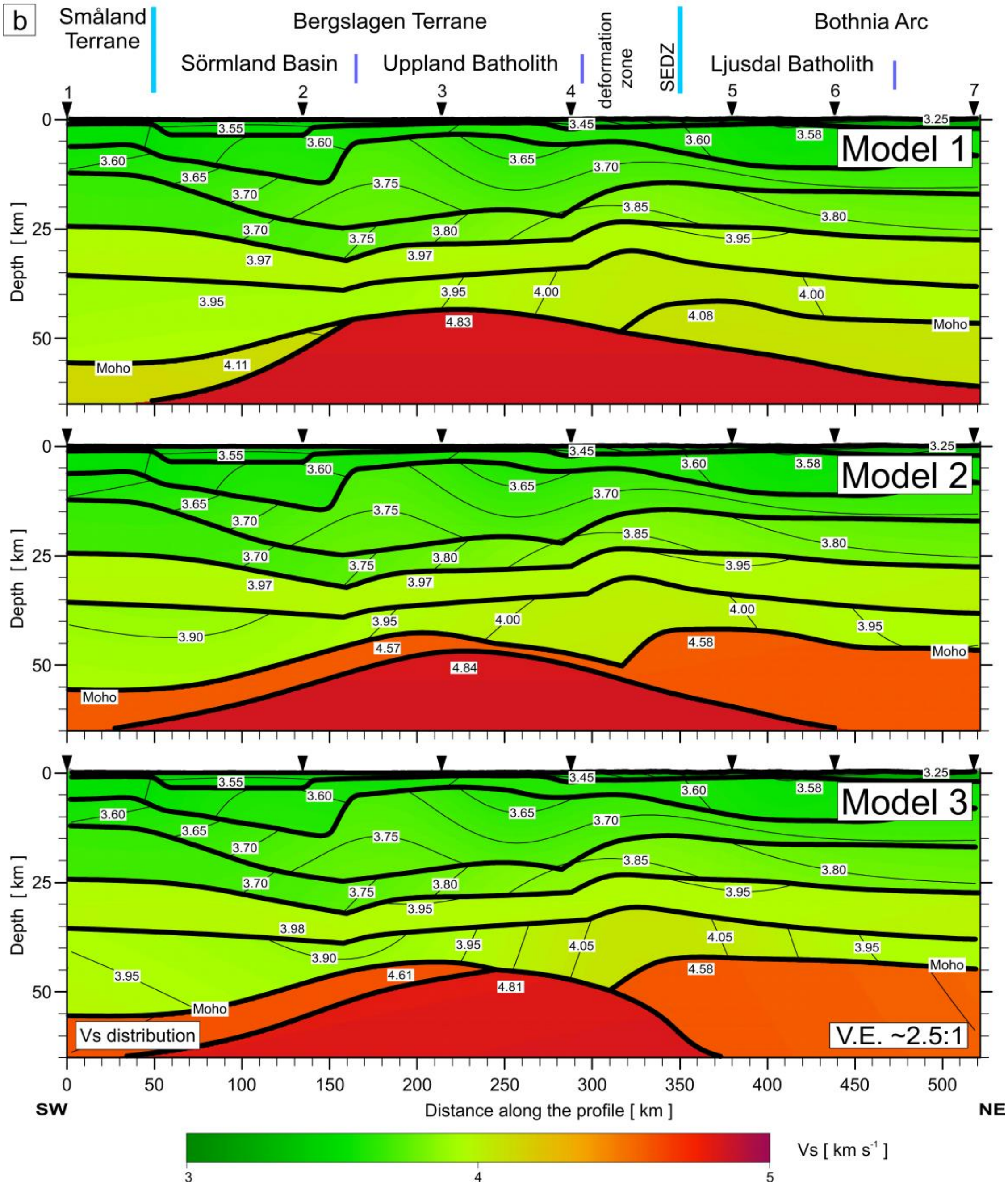

**Figure A4: (continued)**

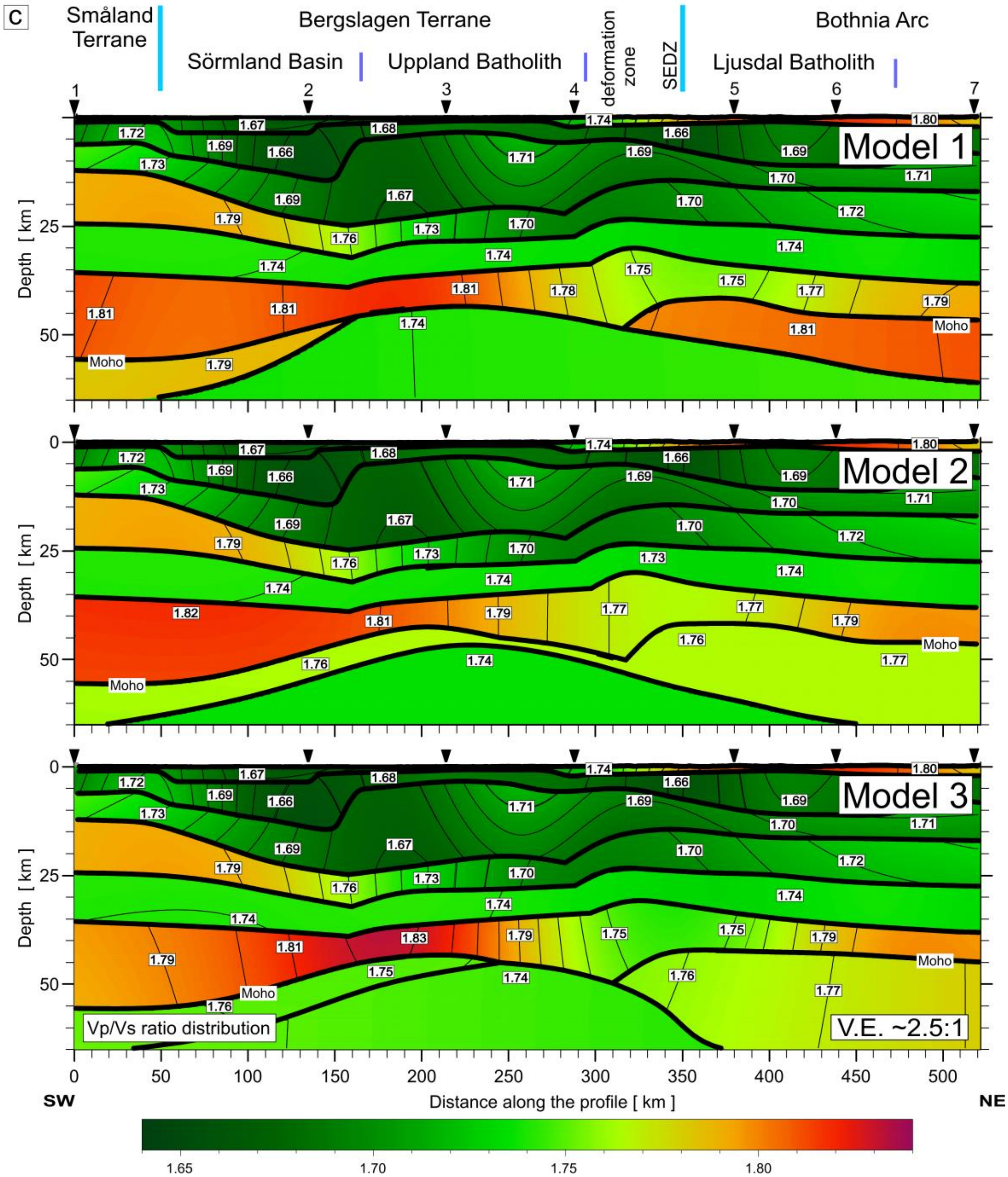

**Figure A4: (continued)**

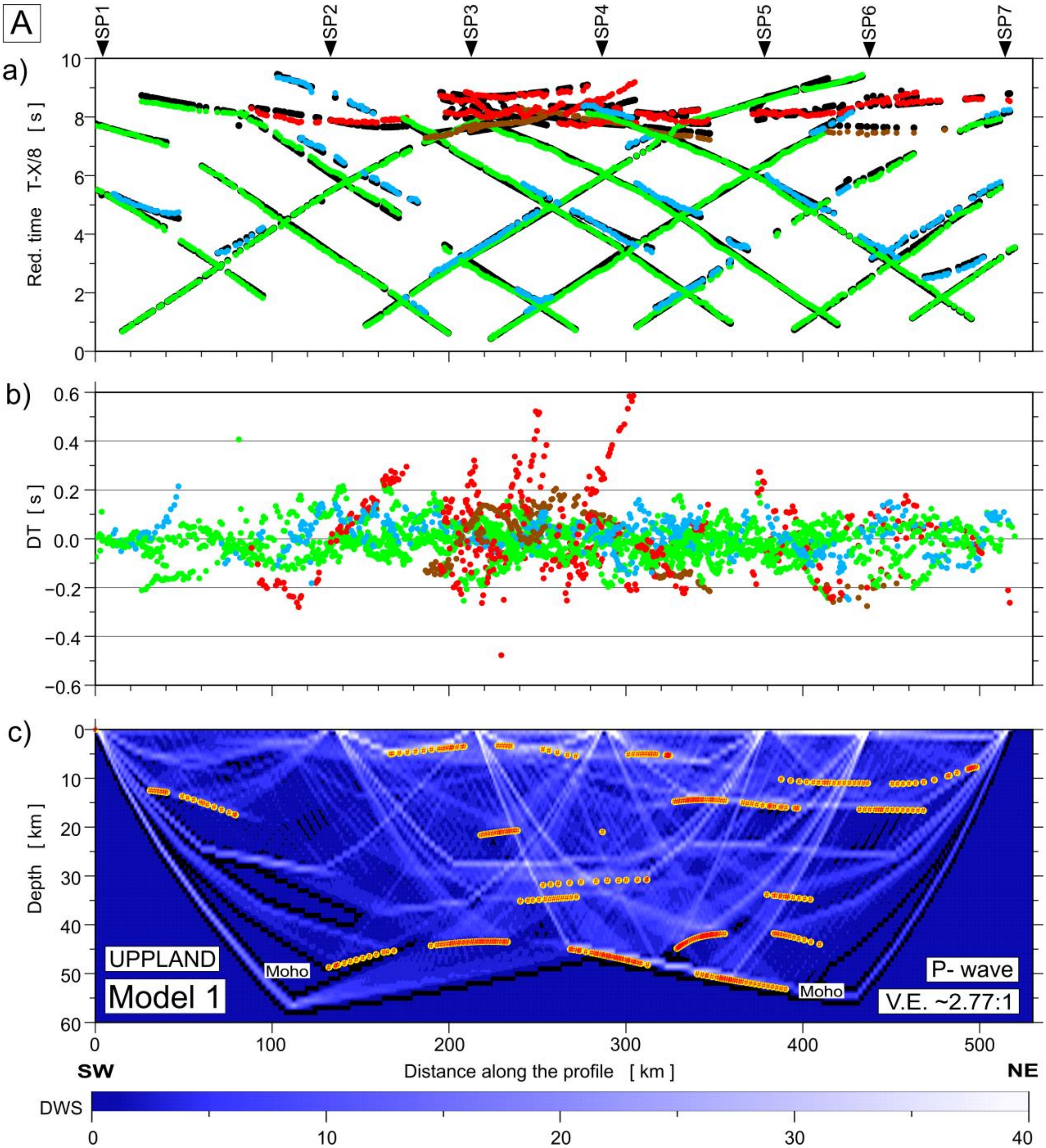

**Figure A5: Diagrams showing theoretical and observed travel times (a), travel time residuals (b), and schematic ray coverage (c) from forward modelling along the profile. Green points – Pg arrivals, blue points – PcP arrivals (reflections in the crust without $P_MP$), red points - $P_MP$, brown points – Pn arrivals, black points – theoretical travel times. Yellow lines – schematic fragments of discontinuities constrained by reflected phases for P-wave velocity Model 1 (A) and (C) for Model 2. The red points plotted along the interfaces mark the theoretical bottoming points of reflected phases (every third point is plotted) and their density is a measure of the positioning accuracy of the reflectors. DWS – derivative**

**weight sum. Respective abbreviations have been used for S-wave for Model 1 (B) and Model 2 (D). The reduction velocity is 8.0 km s-1 for P-wave, and 4.62 km $s^{-1}$ for S-wave.**

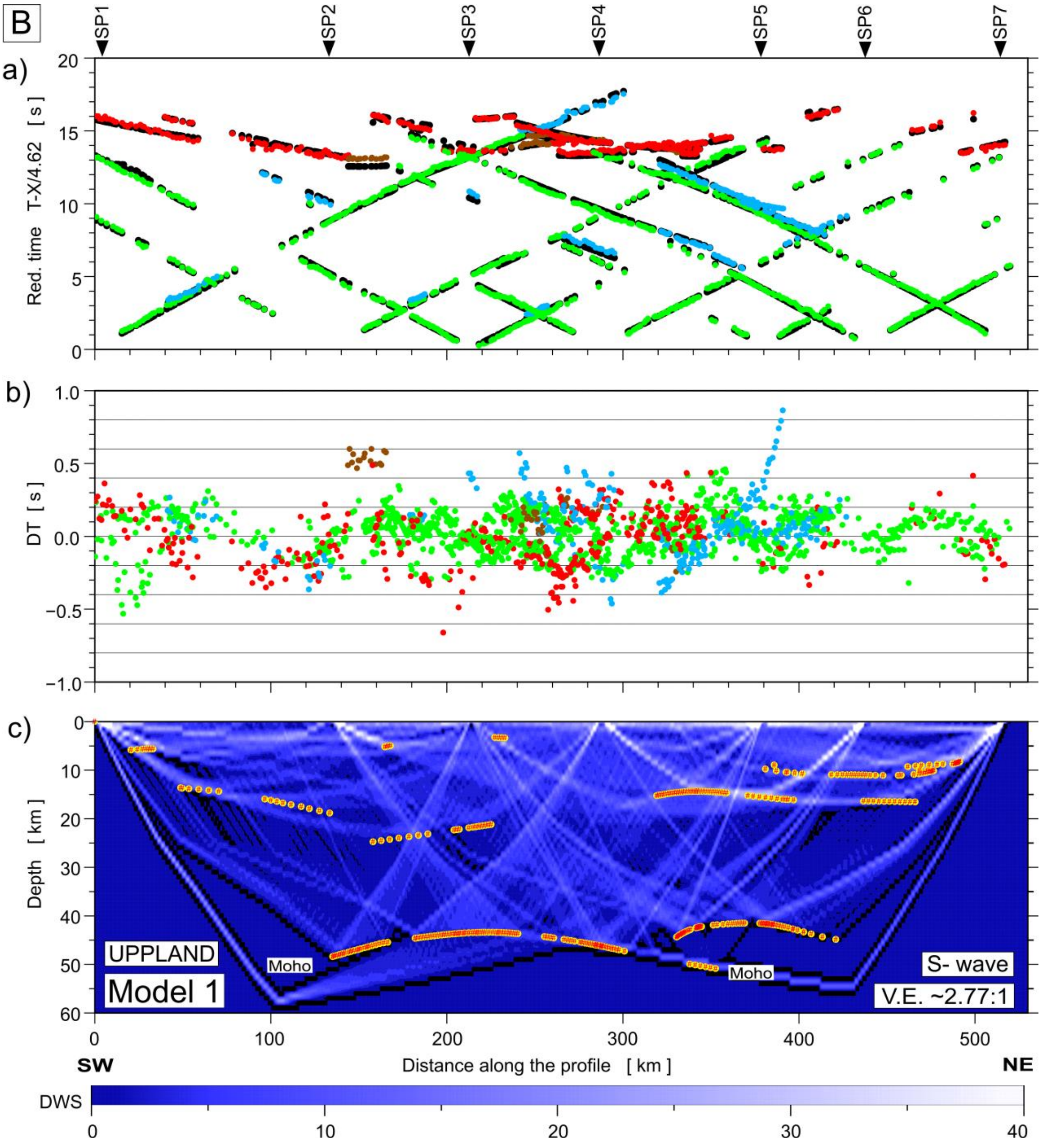

**Figure A5: (continued)**

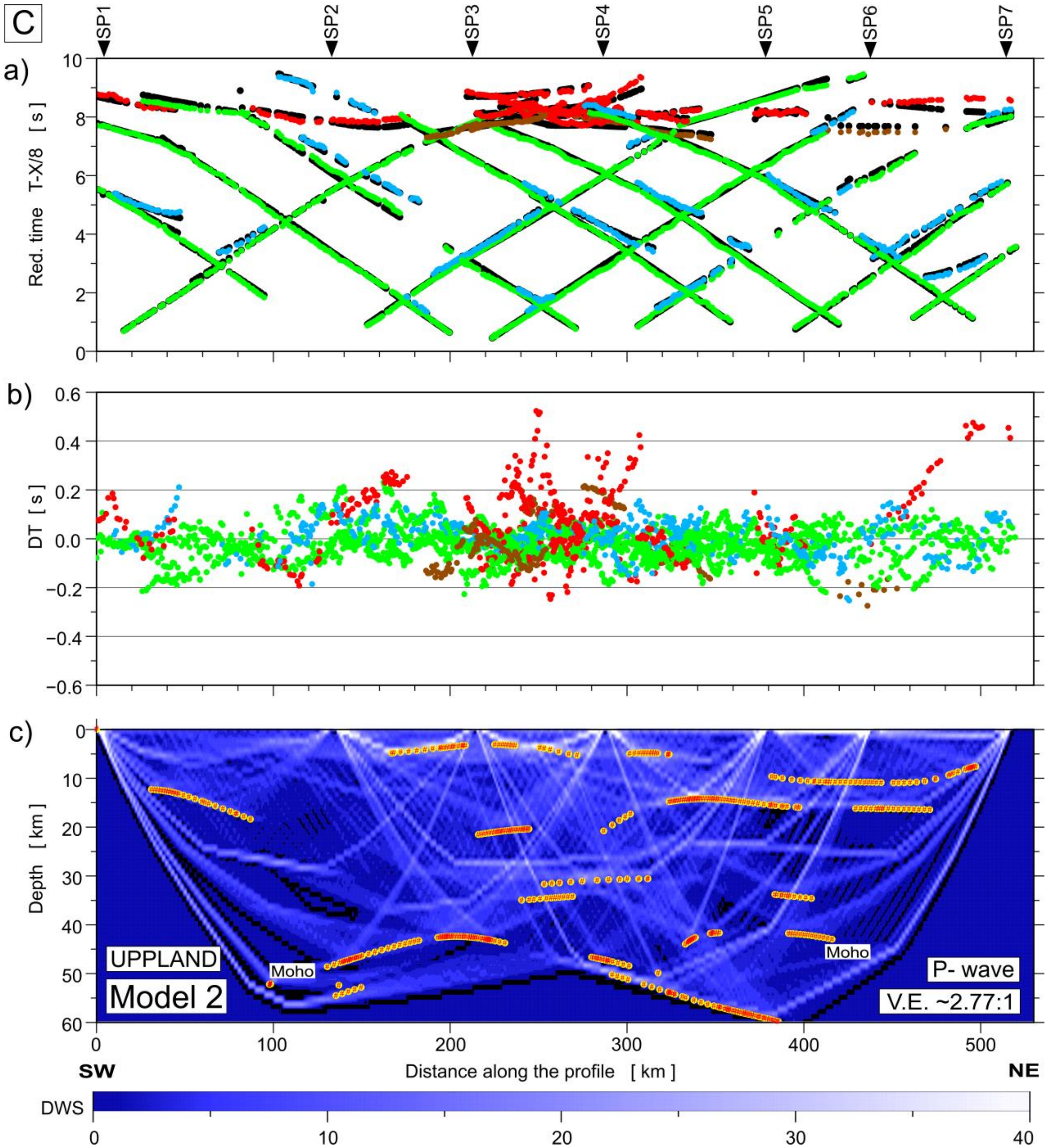

**Figure A5: (continued)**

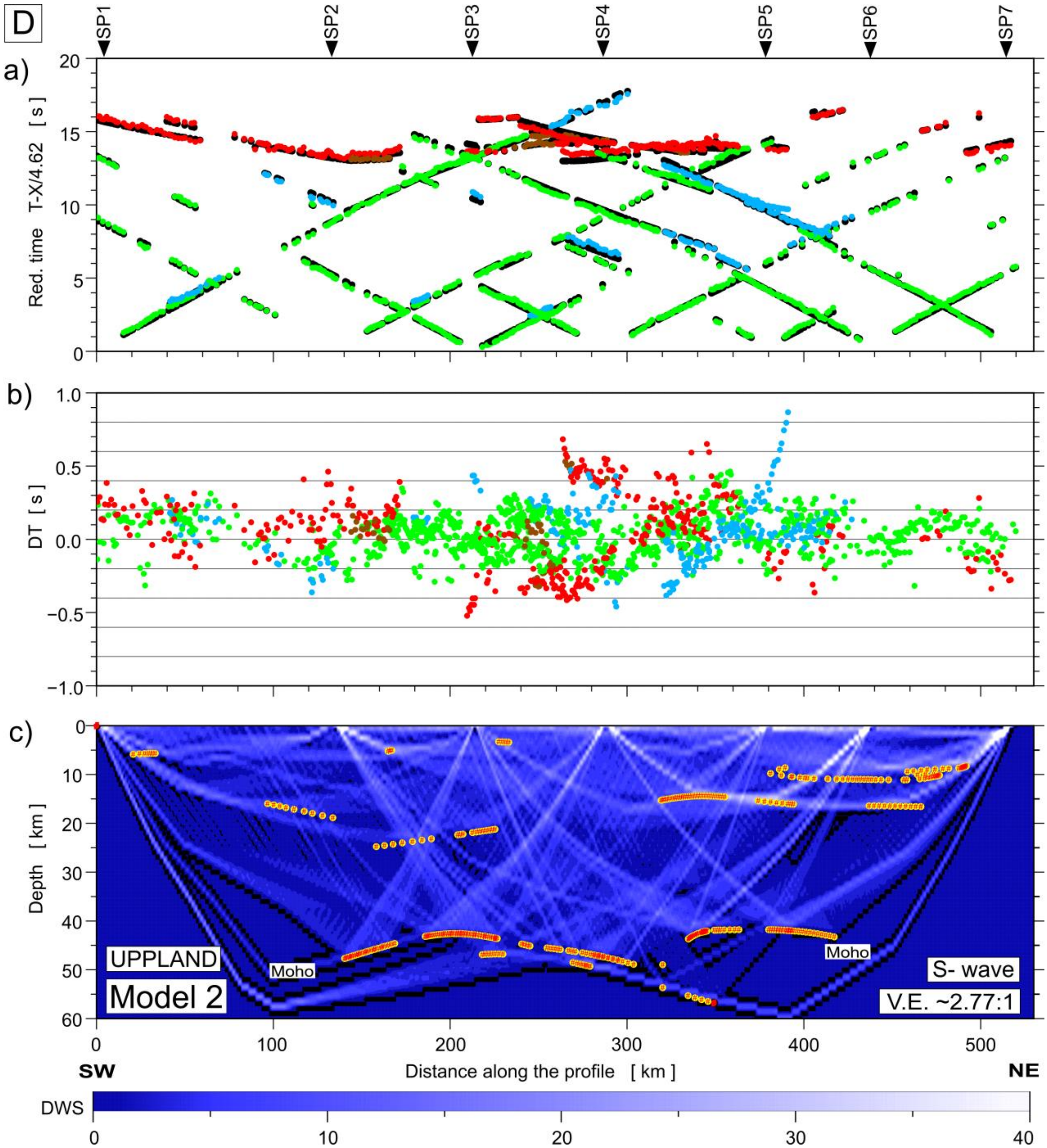

**Figure A5: (continued)**

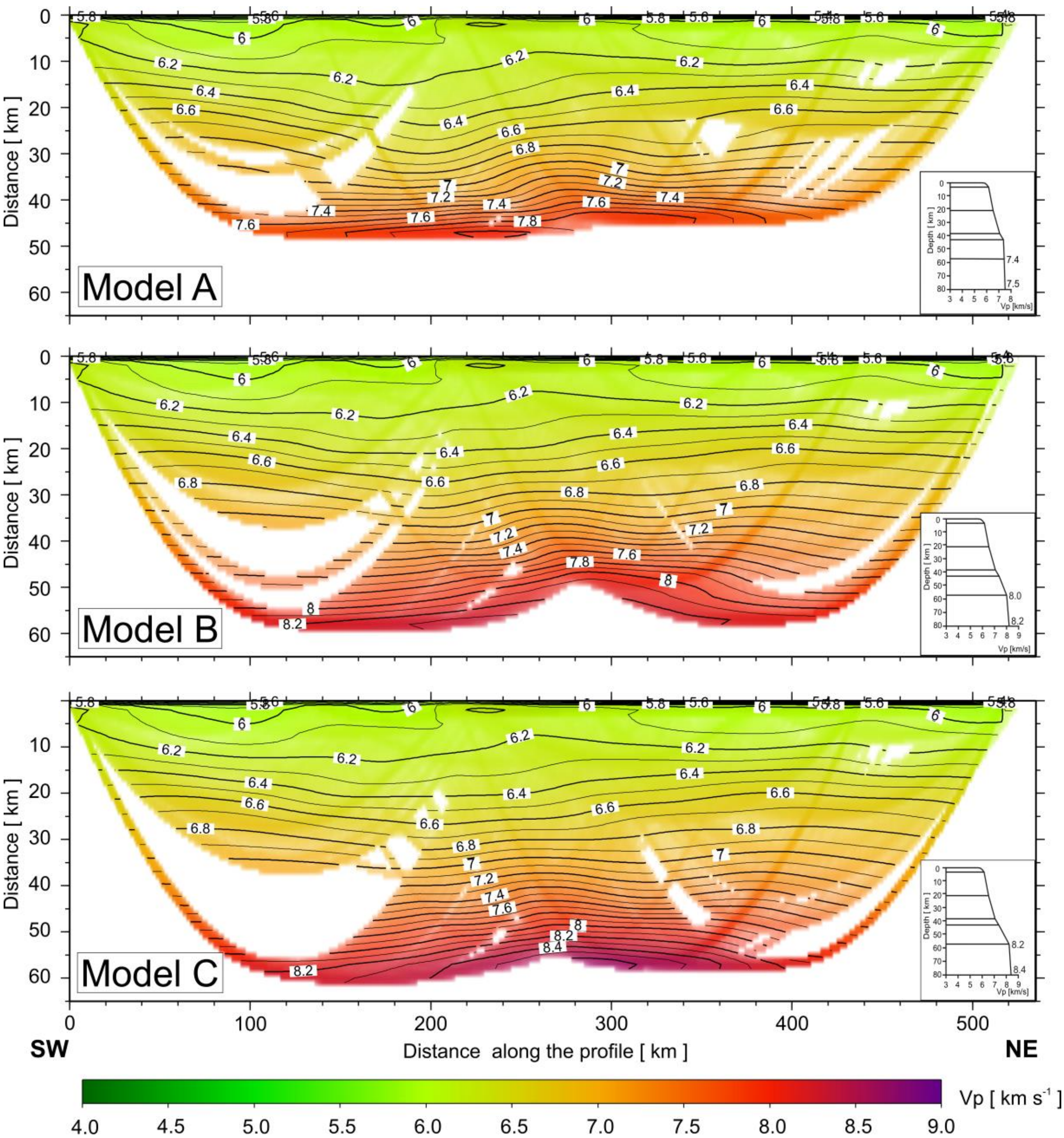

**Figure A6: Results of two-dimensional tomographic inversion of P-wave first arrival travel times, obtained using the program package by Hole (1992). Final 2-D models for different initial 1-D models A, B and C, respectively. Numbers are P-wave velocities in km s$^{-1}$.**

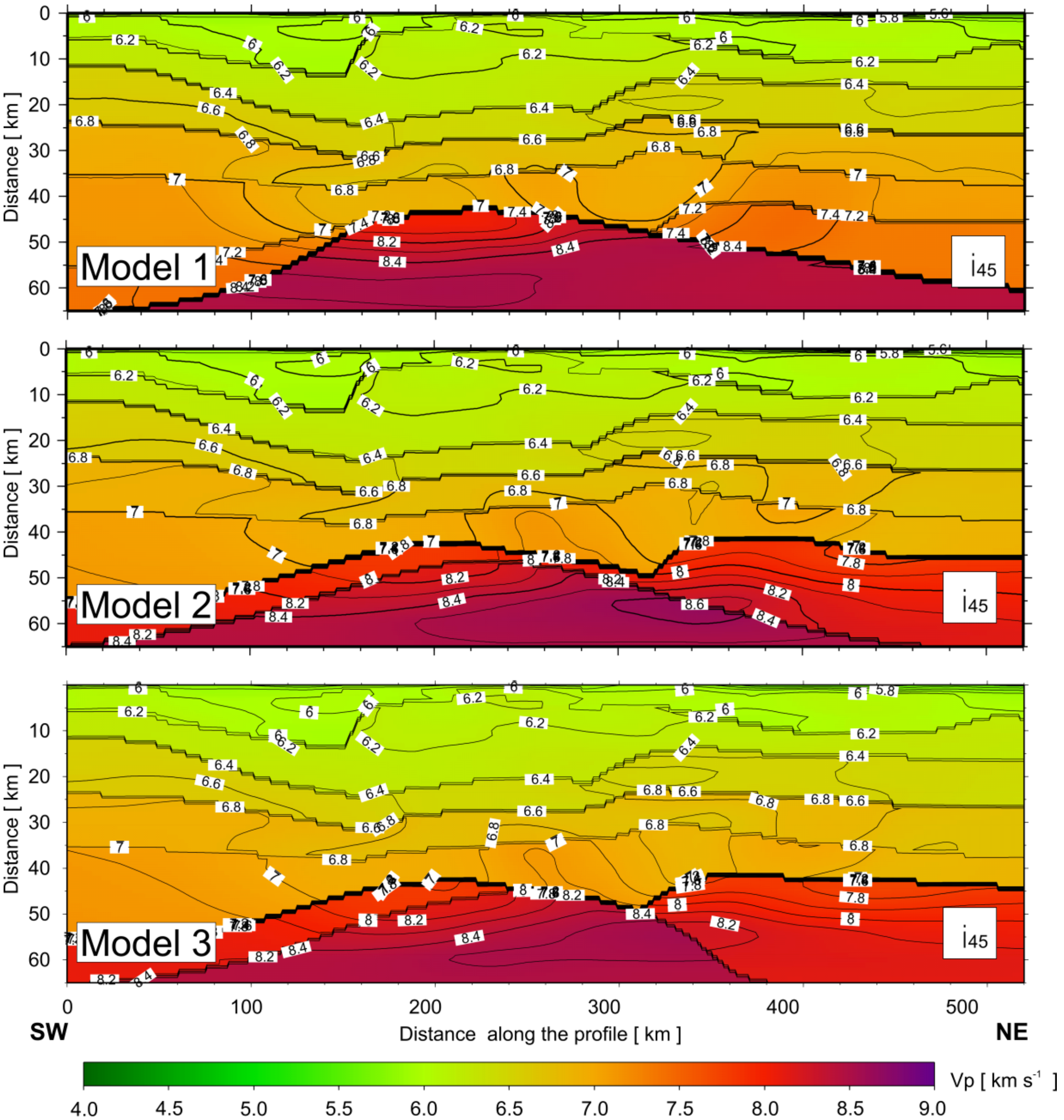

**Figure A7: Results of two-dimensional tomographic inversion of P-wave first arrival travel times, obtained using the program package by Hole (1992). 2-D models with the final velocity fields obtained using Model 1, Model 2, and Model 3 as initial models and rigid boundary geometry after 45 iterations. Numbers are P- wave velocities in km s$^{-1}$.**

**Table A1: Number of picks with RMS values for each P phase for the Model 1.**

| Phase | Number of picks | RMS [s] |
|---|---|---|
| $P_g$ | 2152 | 0.07 |
| $P_n$ | 216 | 0.12 |
| $P_MP$ | 426 | 0.15 |
| $P_{M1}P$ | 74 | 0.23 |
| $P_cP$ | 517 | 0.07 |
| Total | 3385 | 0.09 |

**Table A2: Number of picks with RMS values for each P phase for the Model 2.**

| Phase | Number of picks | RMS [s] |
|---|---|---|
| $P_g$ | 2114 | 0.07 |
| $P_n$ | 35 | 0.13 |
| $P_{n1}$ | 178 | 0.12 |
| $P_MP$ | 348 | 0.13 |
| $P_{M1}P$ | 159 | 0.19 |
| $P_cP$ | 517 | 0.07 |
| Total | 3351 | 0.09 |

**Table A3: Number of picks with RMS values for each P phase for the Model 3.**

| Phase | Number of picks | RMS [s] |
|---|---|---|
| $P_g$ | 2108 | 0.07 |
| $P_n$ | 35 | 0.10 |
| $P_{n1}$ | 181 | 0.11 |
| $P_MP$ | 329 | 0.13 |
| $P_{M1}P$ | 146 | 0.10 |
| $P_cP$ | 513 | 0.07 |
| Total | 3312 | 0.08 |

**Table A4: Number of picks with RMS values for each S phase for the Model 1.**

| Phase | Number of picks | RMS [s] |
|---|---|---|
| $S_g$ | 1099 | 0.28 |
| $S_n$ | 64 | 0.29 |
| $S_MS$ | 380 | 0.18 |
| $S_{M1}S$ | 14 | 0.39 |
| $S_cS$ | 291 | 0.23 |
| Total | 1848 | 0.25 |

**Table A5: Number of picks with RMS values for each S phase for the Model 2.**

| Phase | Number of picks | RMS [s] |
|---|---|---|
| $S_g$ | 1006 | 0.14 |
| $S_n$ | 15 | 0.25 |
| $S_{n1}$ | 49 | 0.21 |
| $S_MS$ | 323 | 0.26 |
| $S_{M1}S$ | 61 | 0.26 |
| $S_cS$ | 291 | 0.23 |
| Total | 1724 | 0.19 |

**Table A6: Number of picks with RMS values for each S phase for the Model 3.**

| Phase | Number of picks | RMS [s] |
|---|---|---|
| $S_g$ | 1067 | 0.14 |
| $S_n$ | 15 | 0.17 |
| $S_{n1}$ | 49 | 0.16 |
| $S_MS$ | 338 | 0.18 |
| $S_{M1}S$ | 41 | 0.17 |
| $S_cS$ | 291 | 0.23 |
| Total | 1780 | 0.16 |

**Table A7: RMS values for the analysed models given for the starting models 1, 2, and 3 after 10 and 45 (Fig. A7 in Appendix A) iterations, respectively**.

| **Iteration** | **Model 1** | **Model 2** | **Model 3** |
|---|---|---|---|
| 0 | 0.18 | 0.13 | 0.20 |
| 10 | 0.08 | 0.07 | 0.08 |
| 15 | 0.05 | 0.05 | 0.05 |