# Peer review of "Revisit of the Fennoscandian Shield along the UPPLAND seismic profile: competitive velocity models"

_EGUsphere, 2025_

## Author Response (AR1)

Dear Editor,

We have replied to both of the reviewers and have corrected the manuscript accordingly (see our ms with highlighted changes). Several figures were also corrected: Fig.1, Fig. 2a, Fig. 2b, Fig. 6, Fig. A1a, Fig. A1b, Fig. A3e, Fig. A4a, Fig. A4b and Fig. A4c.

We are wondering if it is possible to send figures with modeling examples (Fig. 4a, Fig. 4b Fig. A3a, Fig. A3b, Fig. A3c, Fig. A3d and Fig. A3e) in higher resolution. Viewing the modelled details of the seismic sections at the current resolution is problematic. Although the other figures look very good, these could be in better quality.

Taking into account your comments, we have made changes as recommended:

1. We rewrote reference list according to Solid Earth standards.
2. We corrected short summary (it is without abbreviations).
3. We have checked the figures for their appearance for readers with colour vision deficiencies as technical editor suggested. In our opinion, they are legible. Additionally, most of them have values marked on them, which makes them clear enough.
* * *
We would like to thank the reviewer RC#1 - Annakaisa Korja for the review and for valuable comments. We implemented the improvements and corrections to the manuscript as suggested by the reviewer. Below are our detailed answers and explanations concerning all the reviewer's remarks.

**1. Why, have the authors not compared the crustal structure with the subparallel BABEL 6 line to the east. The line is presented in the figure 1. UPPLAND and BABEL 6 seem to share the same Moho architecture between sp4-sp5, when compared without vertical exaggeration.**

We agree, we added suitable comparison to the text. The BABEL 6 line runs almost parallel to the northern part of the UPPLAND profile, at a distance of about 100 km to the east. It should be noted that due to this distance and to the fact that it is a measurement line realized using the seismic reflection method, different from WARR, the comparison of both obtained images of the upper lithospheric structure should be approached with caution. The image along the BABEL 6 line presents a very complicated structure, in which it is difficult to interpret any general boundaries for the entire model. In the middle and lower crust, there are visible separated areas of strong reflections with an extent of several tens of kilometers and a thickness of several kilometers, which are difficult to connect directly. The idea explaining their genesis presented in

the work of Buntin et al., 2019 ("there is a connection between the feeder dyke and up-doming") seems highly probable. The depth of the Moho boundary presented on the BABEL 6 line (Buntin et al., 2019), 50-60 km, is about 5-10 km greater than on the corresponding fragment of our UPPLAND profile model. One of the reasons may be the use of different mean velocities in the crust during processing. The second reason may be the actual change of the Moho depth, which, judging from the European Moho depth map (Grad and Tiira, 2009), tends to increase eastward (by ~3-4 km) from the northern part of the UPPLAND profile to BABEL 6 - as the Reviewer points out in point 4.

**2. When looking at the reflection profiles from the area, it seems that the youngest events are best preserved and thus over-represented in the reflective images of stable crust. Why haven´t the authors described or even referred to the 1.6-1.1 Ga extensional magmatic events that have profoundly changed the reflective image of the crustal structures along BABEL 6,7 and 1 and C profiles. The northern parts of the UPPLAND profile cross-cuts the onshore fringes of the magmatic province from SP4 northwards. This question concerns both maps and text as well as interpretations.**

We agree and have added text to describe younger events, especially the rapakivi-related magmatic events along BABEL 6,7 and 1 and C profiles. Rapakivi-granites have also been added on the map (Fig. 6a). Anyhow, in Bergslagen along the UPPLAND profile, the effects of younger magmatic events on surface geology are nearly non-existent compared to BABEL 6, 7 and 1 and C profiles.

**3. This leads to the third question. Is it possible that some of the crustal and mantle structures are even younger than interpreted? What kind of seismically visible structures would have been either destroyed or enhanced during extensional later events?**

This is an important question and at least the 1.6 Ga rapakivi stage has probably been very important, which possibility was missing in our first interpretation. Otherwise, the 1.2 Ga, 1.0 Ga and other extensional stages can have had some effect, but the magmatic input is minor, and the present data is not detailed enough to solve this. Main problem is that the major bulging of the upper mantle occurs under Bergslagen where we have no evidence for volumetrically important 1.8/1.7/1.6 Ga or younger magmatism. If the Moho is dipping towards east (see below) the mantle bulging could increase towards the west and thus would strengthen more the hypothesis of the 1.8/1.7 Ga stages in the west as cause for bulging compared to the rapakivi stage in the east.

**4. European Moho depth map indicates that UPPLAND profile is located on a Moho slope dipping towards the east. Can this affect the interpretations of the deep reflections observed on UPPLAND profile?**

Very important point and we have used this in our tentative interpretation. If correct it would not favor the 1.6 Ga rapakivi stage in the east as cause for the mantle bulging under the UPPLAND profile. To comment this point from more technical point of view - theoretically, the model for a profile running over the sloping structure of the MOHO boundary has an underestimation of depth because we are modelling depth for rays reflecting perpendicular to the boundary surface. The depth directly below the line of the profile running on the surface is greater. The difference between the two values is greater the steeper the slope of the MOHO surface. In our case, it is relatively small because, according to the MOHO map by Grad et al., 2009, the depth gradient over a distance of just over 100 km between the UPPLAND and BABEL 6 profiles is about 4 km. Consequently, the underestimation of depth will be of the order of tens, at most hundreds of meters, which is much less than the accuracy of our measurements. It is worth recalling that the data from the WARR experiments are the most reliable sources of information for the construction of the MOHO depth map. Slightly less accurate, but providing a picture of the boundary geometry, are data from deep seismic reflection profiles. These two types of data are supplemented, due to the insufficient density of WARR profiles, by less accurate data from passive seismic and gravity surveys.
* * *
We would like to thank the reviewer RC#2 for the review and for valuable comments. We implemented the improvements and corrections to the manuscript as suggested by the reviewer. Below are our detailed answers and explanations concerning all the reviewer's remarks.

**P1 Abstract -> please mention in abstract, which methods were used to constrain the new model**

Done

**P1 L9: was published by Buntin et al. in 2021 -> was published by Buntin et al. (2021)**

Done

**P1 L25-27: …The velocity model (Fig. A2) was calculated, and advanced tectonic and petrological interpretation was also carried out. The great value of the work is comparative litho-geochemistry and velocity analyses for the model … -> would be good to highlight that this concerns the model of Buntin et al. (2021)**

Corrected

**P1 L27: the model, prepared by I. Artiemieva. -> which model is it? Better say e.g., the model of Buntin et al. (2021)**

**Also, I. Artiemieva. -> I. Artemieva.**

Corrected

**P1 L41-43: Verb confusion in this paragraph -> either use future tense or present tense, but systematically when referring to work carried out in this study.**

Corrected

**Fig. 2 caption: seismic record sections for P- and S-wave -> I do not see any S waves**

Corrected

**Fig. 2 caption: Band-pass filters, 2-15 and 1-12 Hz, have been applied, respectively -> it is not clear, which filters to which data were applied, please specify exactly**

Corrected

**Fig. 2 caption: Abbreviations have been used for S-wave, respectively. -> I do not see any S waves in this fig.**

Corrected

**P5 L67: Pmantle = Pn1 -> Two specifications of the same feature is confusing, please use consistently either Pmantle or Pn1, also in seismic sections.**

**Also, do the same for S waves**

Corrected

**P5 L75: $S_n1$ twice -> Sn1 (use either Sn1 or use Smantle consistently, similarly to P waves)**

Corrected

**P8 L107: include the reference for three tested models depicted in Fig. A4**

Done

**P9 L 130, 134: ($P_M1P$) -> ($P_MP$)**

Corrected

**P9 L134: For S-wave, respective residuals are much larger than P wave -> Is it taken into account that also S wave picks have higher uncertainties?**

Yes, the high uncertainties of S picks are the main reason of high S-wave residuals. We added appropriate comment to the text.

**P10 L148, Fig. 5 caption: abbreviations have been used for S-wave -> abbreviations have been used for S-waves**

Done

**P12 L162: first arrivals (Psed + Pg + Pn) -> either explain Psed or remove it Also, I do not find any other mark of Psed, nor it appears in any figure**

Corrected

**P15, Fig. 6: the model in b) is too small while the surface tectonic sketch in a) is quite large -> consider reorganization to enlarge model in panel b).**

**Distinguish lower crust and upper mantle in the model by different colors in Fig. 6b**

**Highlight the Moho in Fig. 6b**

Done

**P27, Fig. A3: section of P waves, confusing labels -> improve labelling for P waves**

Corrected

**P28-30, Fig. A4 a,b,c: Hide unconstrained parts of each model and also in panels a,b,c**

Done

Kind regards,

Monika Bociarska